# Correlation Clustering with Local Objectives

Sanchit Kalhan           Konstantin Makarychev           Timothy Zhou

## Abstract

Correlation Clustering is a powerful graph partitioning model that aims to cluster items based on the notion of similarity between items. An instance of the Correlation Clustering problem consists of a graph $G$ (not necessarily complete) whose edges are labeled by a binary classifier as "similar" and "dissimilar". An objective which has received a lot of attention in literature is that of minimizing the number of disagreements: an edge is in disagreement if it is a "similar" edge and is present across clusters or if it is a "dissimilar" edge and is present within a cluster. Define the disagreements vector to be an $n$ dimensional vector indexed by the vertices, where the $v$-th index is the number of disagreements at vertex $v$. Recently, Puleo and Milenkovic (ICML '16) initiated the study of the Correlation Clustering framework in which the objectives were more general functions of the disagreements vector. In this paper, we study algorithms for minimizing $\ell_q$ norms $(q \geq 1)$ of the disagreements vector for both arbitrary and complete graphs. We present the first known algorithm for minimizing the $\ell_q$ norm of the disagreements vector on arbitrary graphs and also provide an improved algorithm for minimizing the $\ell_q$ norm $(q \geq 1)$ of the disagreements vector on complete graphs. We also study an alternate cluster-wise local objective introduced by Ahmadi, Khuller and Saha (IPCO '19), which aims to minimize the maximum number of disagreements associated with a cluster. We also present an improved $(2 + \varepsilon)$-approximation algorithm for this objective. Finally, we compliment our algorithmic results for minimizing the $\ell_q$ norm of the disagreements vector with some hardness results.

## 1   Introduction

A basic task in machine learning is that of clustering items based on the similarity between them. This task can be elegantly captured by Correlation Clustering, a clustering framework first introduced by Bansal et al. [2004]. In this model, we are given access to items and the *similarity/dissimilarity* between them in the form of a graph $G$ on $n$ vertices. The edges of $G$ represent whether the items are *similar* or *dissimilar* and are labelled as ("+") and ("−") respectively. The goal is to produce a clustering that agrees with the labeling of the edges as much as possible, i.e., to group positive edges in the same cluster and place negative edges across different clusters (a positive edge that is present across clusters or a negative edge that is present within the same cluster is said to be in disagreement). The Correlation Clustering problem can be viewed as an agnostic learning problem, where we are given noisy examples and the task is to fit a hypothesis as best as possible to these examples. Co-reference resolution (see e.g., Cohen and Richman [2001, 2002]), spam detection (see e.g., Ramachandran et al. [2007], Bonchi et al. [2014]) and image segmentation (see e.g., Wirth [2017]) are some of the applications to which Correlation Clustering has been applied to in practice.

This task is made trivial if the labeling given is consistent (transitive): if $(u, v)$ and $(v, w)$ are similar, then $(u, w)$ is similar for all vertices $u, v, w$ in $G$ (the connected components on similar edges would give an optimal clustering). Instead, it is assumed that the given labeling is inconsistent, i.e., it is possible that $(u, w)$ are dissimilar even though $(u, v)$ and $(v, w)$ are similar. For such a triplet $u, v, w$, every possible clustering incurs a disagreement on at least one edge and thus, no perfect clustering

exists. The optimal clustering is the one which minimizes the disagreements. Moreover, as the number of clusters is not predefined, the optimal clustering can use anywhere from 1 to $n$ clusters.

Minimizing the total weight of edges in disagreement is the objective that has received the most consideration in literature. Define the disagreements vector be an $n$ dimensional vector indexed by the vertices where the $v$-th coordinate equals the number of disagreements at $v$. Thus, minimizing the total number of disagreements is equivalent to minimizing the $\ell_1$ norm of the disagreements vector. Puleo and Milenkovic [2016] initiated the study of local objectives in the Correlation Clustering framework. They focus on complete graphs and study the minimization of $\ell_q$ norms ($q \geq 1$) of the disagreements vector – for which they provided a $48-$approximation algorithm. Charikar, Gupta, and Schwartz [2017] gave an improved $7-$approximation algorithm for minimizing $\ell_q$ disagreements on complete graphs. They also studied the problem of minimizing the $\ell_\infty$ norm of the disagreements vector (also known as Min Max Correlation Clustering) for arbitrary graphs, for which they provided a $O(\sqrt{n})-$approximation.

For higher values of $q$ (particularly $q = \infty$), a clustering optimized for minimizing the $\ell_q$ norm prioritizes reducing the disagreements at vertices that are worst off. Thus, such metrics are very unforgiving in most cases as it is possible that in the optimal clustering there is only one vertex with high disagreements while every other vertex has low disagreements. Hence, one is forced to infer the most pessimistic picture about the overall clustering. The $\ell_2$ norm is a solution to this tension between the $\ell_1$ and $\ell_\infty$ objectives. The $\ell_2$ norm of the disagreements vector takes into account the disagreements at each vertex while also penalizing the vertices with high disagreements more heavily. Thus, a clustering optimized for the minimum $\ell_2$ norm gives a more balanced clustering as it takes into consideration both the global and local picture.

Recently, Ahmadi, Khuller, and Saha [2019b] introduced an alternative min max objective for correlation clustering (which we call AKS min max objective). For a cluster $C \subseteq V$, let us refer to similar edges with exactly one endpoint in $C$ and dissimilar edges with both endpoints in $C$ as edges in disagreements with respect to $C$. We call the weight of all edges in disagreement with $C$ the cost of $C$. Then, the AKS min max objective asks to find a clustering $C_1, \ldots, C_T$ that minimizes the maximum cost $C_i$. Ahmadi et al. [2019b] gave an $O(\log n)-$approximation algorithm for this objective. Ahmadi, Galhotra, Khuller, Saha, and Schwartz [2019a] improved the approximation factor to $O(\sqrt{\log n} \cdot \max\{\log |E^-|, \log(k)\})$.

**Our contributions.** In this paper, we provide positive and negative results for Correlation Clustering with the $\ell_q$ objective. We first study the problem of minimizing disagreements on arbitrary graphs. We present the first approximation algorithm minimizing any $\ell_q$ norm ($q \geq 1$) of the disagreements vector.

**Theorem 1.1.** *There exists a polynomial time $O(n^{\frac{1}{2}-\frac{1}{2q}} \cdot \log^{\frac{1}{2}+\frac{1}{2q}} n)-$approximation algorithm for the minimum $\ell_q$ disagreements problem on general weighted graphs.*

For the $\ell_2$ objective, the above algorithm leads to an approximation ratio of $\tilde{O}(n^{1/4})$, thus providing the first known approximation ratio for optimizing the clustering for this version of the objective. Note that the above algorithm matches the best approximation guarantee of $O(\log n)$ for the classical objective of minimizing the $\ell_1$ norm of the disagreements vector. For the $\ell_\infty$ norm, our algorithm matches the guarantee of the algorithm by Charikar, Gupta, and Schwartz [2017] up to log factors. Fundamental combinatorial optimization problems like *Multicut, Multiway Cut* and *s-t Cut* can be framed as special cases of Correlation Clustering. Thus, Theorem 1.1 leads to the first known algorithms for *Multicut, Multiway Cut* and *s-t Cut* with the $\ell_q$ objective when $q \neq 1$ and $q \neq \infty$. We can also use the algorithm from Theorem 1.1 to obtain $O(n^{\frac{1}{2}-\frac{1}{2q}} \cdot \log^{\frac{1}{2}+\frac{1}{2q}} n)$ bi-criteria approximation for Min $k$-Balanced Partitioning with the $\ell_q$ objective (we omit details here).

Next, we study the case of complete graphs. For this case, we present an improved $5-$approximation algorithm for minimizing any $\ell_q$ norm ($q \geq 1$) of the disagreements vector.

**Theorem 1.2.** *There exists a polynomial time $5-$approximation algorithm for the minimum $\ell_q$ disagreements problem on complete graphs.*

We also study the case of complete bipartite graphs where disagreements need to be bounded for only one side of the bipartition, and not the whole vertex set. We give an improved $5-$approximation algorithm for minimizing any $\ell_q$ norm ($q \geq 1$) of the disagreements vector.

**Theorem 1.3.** *There exists a polynomial time* $5-$*approximation algorithm for the minimum* $\ell_q$ *disagreements problem on complete bipartite graphs where disagreements are measured for only one side of the bipartition.*

In this paper, we also consider the AKS min max objective. For this objective, we give a $(2 + \varepsilon)-$approximation algorithm, which improves the approximation ratio of $O(\sqrt{\log n} \cdot \max\{\log |E^-|, \log(k)\})$ given by Ahmadi, Galhotra, Khuller, Saha, and Schwartz [2019a].

**Theorem 1.4.** *There exists a polynomial time* $(2 + \varepsilon)-$*approximation algorithm for the AKS min max problem on arbitrary graphs.*

Finally, in the full version of this paper (see supplemental materials), we present an integrality gap of $\Omega(n^{\frac{1}{2} - \frac{1}{2q}})$ for minimum $\ell_q$ $s - t$ cut and prove a hardness of approximation of 2 for minimum $\ell_\infty$ $s - t$ cut.

**Previous work.** Bansal, Blum, and Chawla [2004] showed that it is NP-hard to find a clustering that minimizes the total disagreements, even on complete graphs. They give a constant-factor approximation algorithm to minimize disagreements and a PTAS to maximize agreements on complete graphs. For complete graphs, Ailon, Charikar, and Newman [2008] presented a randomized algorithm with an approximation guarantee of 3 to minimize total disagreements. They also gave a 2.5 approximation algorithm based on LP rounding. This factor was improved to slightly less than 2.06 by Chawla, Makarychev, Schramm, and Yaroslavtsev [2015]. Since, the natural LP is known to have an integrality gap of 2, the problem of optimizing the classical objective is almost settled with respect to the natural LP. For arbitrary graphs, the best known approximation ratio is $O(\log n)$ (see Charikar, Guruswami, and Wirth [2003], Demaine, Emanuel, Fiat, and Immorlica [2006]). Assuming the Unique Games Conjecture, there is no constant-factor approximation algorithm for minimizing $\ell_1$ disagreements on arbitrary graphs (see Chawla et al. [2006]). Puleo and Milenkovic [2016] first studied Correlation Clustering with more local objectives. For minimizing $\ell_q$ $(q \geq 1)$ norms of the disagreements vector on complete graphs, their algorithm achieves an approximation guarantee of 48. This was improved to 7 by Charikar, Gupta, and Schwartz [2017]. Charikar et al. [2017] also studied the problem of minimizing the $\ell_\infty$ norm of the disagreements vector on general graphs. They showed that the natural LP/SDP has an integrality gap of $n/2$ for this problem and provided a $O(\sqrt{n})-$approximation algorithm for minimum $\ell_\infty$ disagreements. Puleo and Milenkovic [2016] also initiated the study of minimizing the $\ell_q$ norm of the disagreements vector (for one side of the bipartition) on complete bipartite graphs. The presented a $10-$approximation algorithm for this problem, which was improved to 7 by Charikar, Gupta, and Schwartz [2017]. Recently, Ahmadi et al. [2019b] studied an alternative objective for the correlation clustering problem. Motivated by creating balanced communities for problems such as image segmentation and community detection in social networks, they propose a new cluster-wise min-max objective. This objective minimizes the maximum weight of edges in disagreement associated with a cluster, where an edge is in disagreement with respect to a cluster if it is a similar edge and has exactly one end point in the cluster or if it is a dissimilar edge and has both its endpoints in the cluster. They gave an $O(\sqrt{\log n} \cdot \max\{\log |E^-|, \log(k)\})-$approximation algorithm for this objective. Moreover, they give a $O(r^2)-$approximation algorithm for graphs that exclude a $K_{r,r}$ minor, and a $14-$approximation algorithm for complete graphs.

## 2 Preliminaries

We now formally define the Correlation Clustering with $\ell_q$ objective problem. We will need the following definition. Consider a set of points $V$ and two disjoint sets of edges on $V$: positive edges $E^+$ and negative edges $E^-$. We assume that every edge has a weight $w_{uv}$. For every partition $\mathcal{P}$ of $V$, we say that a positive edge is in disagreement with $\mathcal{P}$ if the endpoints $u$ and $v$ belongs to different parts of $\mathcal{P}$; and a negative edge is in disagreement with $\mathcal{P}$ if the endpoints $u$ and $v$ belongs to the same part of $\mathcal{P}$. The vector of disagreements, denoted by $\text{disagree}(\mathcal{P}, E^+, E^-)$, is a $|V|$ dimensional vector indexed by elements of $V$. Its coordinate $u$ equals

$$\text{disagree}_u(\mathcal{P}, E^+, E^-) = \sum_{v:(u,v)\in E^+\cup E^-} w_{uv} \mathbb{1}((u,v) \text{ is in disagreement with } \mathcal{P}).$$

$$\text{minimize} \quad \max\left(\|y\|_q, \left(\sum_{u \in V} z_u\right)^{\frac{1}{q}}\right) \tag{P}$$

$$\text{subject to} \quad y_u = \sum_{v:(u,v)\in E^+} w_{uv} x_{uv} + \sum_{v:(u,v)\in E^-} w_{uv}(1 - x_{uv}) \quad \text{for all } u \in V \tag{P1}$$

$$z_u = \sum_{v:(u,v)\in E^+} w_{uv}^q x_{uv} + \sum_{v:(u,v)\in E^-} w_{uv}^q (1 - x_{uv}) \quad \text{for all } u \in V \tag{P2}$$

$$x_{v_1 v_2} + x_{v_2 v_3} \geq x_{v_1 v_3} \qquad\qquad \text{for all } v_1, v_2, v_3 \in V \tag{P3}$$
$$x_{uv} = x_{vu} \qquad\qquad \text{for all } u, v \in V \tag{P4}$$
$$x_{uv} \in [0, 1] \qquad\qquad \text{for all } u, v \in V \tag{P5}$$

Figure 3.1: Convex relaxation for Correlation Clustering with min $\ell_q$ objective for $q < \infty$.

That is, $\text{disagree}_u(\mathcal{P}, E^+, E^-)$ is the weight of disagreeing edges incident to $u$. We similarly define a cut vector for a set of edges $E$:

$$\text{cut}_u(\mathcal{P}, E) = \sum_{v:(u,v)\in E} w_{uv} \mathbb{1}(u \text{ and } v \text{ are separated by } \mathcal{P}).$$

We use the standard definition for the $\ell_q$ norm of a vector $x$: $\|x\|_q = \left(\sum_u x_u^q\right)^{\frac{1}{q}}$ and $\|x\|_\infty = \max_u x_u$. For a partition $\mathcal{P}$, we denote by $\mathcal{P}(u)$ the piece that contains vertex $u$.

**Definition 1.** *In the Correlation Clustering problem with $\ell_q$ objective, we are given a graph $G$ on a set $V$ with two disjoint sets of edges $E^+$ and $E^-$ and a set of weights $w_{uv}$. The goal is find a partition $\mathcal{P}$ that minimizes the $\ell_q$ norm of the disagreements vector, $\|\text{disagree}(\mathcal{P}, E^+, E^-)\|_q$.*

In our algorithm for Correlation Clustering on arbitrary graphs, we will use a powerful technique of padded metric space decompositions (see e.g., Bartal [1996], Rao [1999], Fakcharoenphol and Talwar [2003], Gupta, Krauthgamer, and Lee [2003]).

**Definition 2** (Padded Decomposition). *Let $(X, d)$ be a metric space on $n$ points, and let $\Delta > 0$. A probabilistic distribution of partitions $\mathcal{P}$ of $X$ is called a padded decomposition if it satisfies the following properties:*

- *Each cluster $C \in \mathcal{P}$ has diameter at most $\Delta$.*

- *For every $u \in X$ and $\varepsilon > 0$, $\Pr(\text{Ball}(u, \delta) \not\subset \mathcal{P}(u)) \leq D \cdot \frac{\delta}{\Delta}$ where $\text{Ball}(u, \delta) = \{v \in X : d(u,v) \leq \delta\}$*

**Theorem 2.1** (Fakcharoenphol, Rao, and Talwar [2004]). *Every metric space $(X, d)$ on $n$ points admits a $D = O(\log n)$ separating padded decomposition. Moreover, there is a polynomial-time algorithm that samples a partition from this distribution.*

# 3 Convex Relaxation

In our algorithms for minimizing $\ell_q$ disagreements in arbitrary and complete graphs, we use a convex relaxation given in Figure 3.1. Our convex relaxation for Correlation Clustering is fairly standard. It is similar to relaxations used in the papers by Garg, Vazirani, and Yannakakis [1996], Demaine, Emanuel, Fiat, and Immorlica [2006], Charikar, Guruswami, and Wirth [2003]. For every pair of vertices $u$ and $v$, we have a variable $x_{uv}$ that is equal to the distance between $u$ and $v$ in the "multicut metric". Variables $x_{uv}$ satisfy the triangle inequality constraints (P3). They are also symmetric (P4) and $x_{uv} \in [0, 1]$ (P5). Thus, the set of vertices $V$ equipped with the distance function $d(u, v) = x_{uv}$ is a metric space.

Additionally, for every vertex $u \in V$, we have variables $y_u$ and $z_u$ (see constraints (P1) and (P2)) that lower bound the number of disagreeing edges incident to $u$. The objective of our convex program is to minimize $\max(\|y\|_q, (\sum_u z_u)^{\frac{1}{q}})$. Note that all constraints in the program (P) are linear; however, the objective function of (P) is not convex as is. So in order to find the optimal solution, we

raise the objective function to the power of $q$ and find feasible $x, y, z$ that minimizes the objective $\max(\|y\|_q^q, \sum_u z_u)$.

This program has a polynomial number of linear constraints, and its objective function is convex: This is because the objective function, $\max(\|y\|_q^q, \sum_u z_u)$, is the maximum of two convex functions. The first function, $\|y\|_q^q$ is the sum of $q$-th powers of the variables $y_u$ which are positive. Thus, $\|y\|_q^q$ is convex and differentiable. The second function, $\sum_u z_u$ is a linear function. Therefore, we can use off-the-shelf convex solvers (quadratic solvers for $\ell_2$) to get an optimal solution to $(P)$.

Let us verify that program (P) is a relaxation for Correlation Clustering. Consider an arbitrary partitioning $\mathcal{P}$ of $V$. In the integral solution corresponding to $\mathcal{P}$, we set $x_{uv} = 0$ if $u$ and $v$ are in the same cluster in $\mathcal{P}$; and $x_{uv} = 1$ if $u$ and $v$ are in different clusters in $\mathcal{P}$. In this solution, distances $x_{uv}$ satisfy triangle inequality constraints (P3) and $x_{uv} = x_{vu}$ (P4). Observe that a positive edge $(u, v) \in E^+$ is in disagreement with $\mathcal{P}$ if $x_{uv} = 1$; and a negative edge $(u, v) \in E^-$ is in disagreement if $x_{uv} = 0$. Thus, in this integral solution, $y_u = \text{disagree}_u(\mathcal{P}, E^+, E^-)$ and moreover, $z_u \leq y_u^q$. Therefore, in the integral solution corresponding to $\mathcal{P}$, the objective function of (P) equals $\|\text{disagree}_u(\mathcal{P}, E^+, E^-)\|_q$. Of course, the cost of the optimal fractional solution to the problem may be less than the cost of the optimal integral solution. Thus, (P) is a relaxation for our problem. Below, we denote the cost of the optimal fraction solution to (P) by $LP$.

We remark that we can get a simpler relaxation by removing variables $z$ and changing the objective function to $\|y\|_q$. This relaxation also works for $\ell_\infty$ norm. We use it in our 5-approximation algorithm.

# 4  Overview of Algorithms

We note that some proofs from Subsections 4.1, 4.2 and 4.3 have been deferred to Sections A, B and C respectively (in the supplementary material). These Lemmas and their proofs have been referrenced appropriately.

## 4.1  Correlation Clustering on arbitrary graphs

In this section, we describe our algorithm for minimizing $\ell_q$ disagreements on arbitrary graphs. We will prove the following main theorem.

**Theorem 4.1.** *There exists a randomized polynomial-time $O(n^{\frac{q-1}{2q}} \log^{\frac{q+1}{2q}} n)$−approximation algorithm for Correlation Clustering with the $\ell_q$ objective ($q \geq 1$).*

We remark that the same algorithm gives $O(\sqrt{n \log n})$−approximation for the $\ell_\infty$ norm. We omit the details in the conference version of the paper.

Our algorithm relies on a procedure for partitioning arbitrary metric spaces into pieces of small diameter. In particular, we prove the following theorem,

**Theorem 4.2.** *There exists a polynomial-time randomized algorithm that given a metric space $(X, d)$ on $n$ points and parameter $\Delta$ returns a random partition $\mathcal{P}$ of $X$ such that the diameter of every set $P$ in $\mathcal{P}$ is at most $\Delta$ and for every $q \geq 1$ ($q \neq \infty$) and every weighted graph $G = (X, E, w)$, we have*

$$\mathbb{E}\Big[\|\operatorname{cut}(\mathcal{P}, E)\|_q\Big] \leq Cn^{\frac{q-1}{2q}} \log^{\frac{q+1}{2q}} n \cdot \Big[\Big(\sum_{u \in X} \sum_{v:(u,v)\in E} w_{uv}^q \frac{d(u,v)}{\Delta}\Big)^{1/q} +$$
$$+ \Big(\sum_{u \in X} \Big(\sum_{v:(u,v)\in E} w_{uv} \frac{d(u,v)}{\Delta}\Big)^q\Big)^{1/q}\Big], \quad (1)$$

*for some absolute constant $C$.*

We defer the proof of the above theorem to Section A.

We now show how to use the above metric space partitioning scheme to obtain an approximation algorithm for Correlation Clustering. Note that this proves Theorem 4.1.

*Proof of Theorem 4.1.* Our algorithm first finds the optimal solution $x, y, z$ to the convex relaxation (P) presented in Section 3. Then, it defines a metric $d(u, v) = x_{uv}$ on the vertices of the graph. Finally, it runs the metric space partitioning algorithm with $\Delta = 1/2$ from Section A (see Theorem 4.2) and outputs the obtained partitioning $\mathcal{P}$.

Let us analyze the performance of this algorithm. Denote the cost of the optimal solution $x, y, z$ by $LP$. We know that the cost of the optimal solution $OPT$ is lower bounded by $LP$ (see Section 3 for details). By Theorem 4.2, applied to the graph $G = (V, E^+)$ (note: we ignore negative edges for now),

$$\mathbb{E}\left[\|\operatorname{cut}(\mathcal{P}, E^+)\|_q\right] \leq \frac{C}{\Delta} n^{\frac{q-1}{2q}} \log^{\frac{q+1}{2q}} n \cdot \left(\left(\sum_{u \in V} y_u^q\right)^{\frac{1}{q}} + \left(\sum_{u \in V} z_u\right)^{\frac{1}{q}}\right) \leq 4Cn^{\frac{q-1}{2q}} \log^{\frac{q+1}{2q}} n \cdot LP.$$
(2)

Recall that a positive edge is not in agreement if and only if it is cut. Hence, $\operatorname{disagree}(\mathcal{P}, E^+, \varnothing) = \operatorname{cut}(\mathcal{P}, E^+)$, and the bound above holds for $\mathbb{E}\|\operatorname{disagree}(\mathcal{P}, E^+, \varnothing)\|_q$. By the triangle inequality, $\mathbb{E}\|\operatorname{disagree}(\mathcal{P}, E^+, E^-)\|_q \leq \mathbb{E}\|\operatorname{disagree}(\mathcal{P}, E^+, \varnothing)\|_q + \mathbb{E}\|\operatorname{disagree}(\mathcal{P}, \varnothing, E^-)\|_q$. Hence, to finish the proof, it remains to upper bound $\mathbb{E}\|\operatorname{disagree}(\mathcal{P}, \varnothing, E^-)\|_q$.

Observe that the diameter of every cluster returned by the algorithm is at most $\Delta = 1/2$. For all disagreeing negative edges $(u, v) \in E^-$, we have $x_{uv} \leq 1/2$ and $1 - x_{uv} \geq 1/2$. Thus, $\operatorname{disagree}_u(\mathcal{P}, \varnothing, E^-) \leq 2y_u$ for every $u$, and $\mathbb{E}\|\operatorname{disagree}(\mathcal{P}, \varnothing, E^-)\|_q \leq 2\|y\|_q \leq 2LP$. This completes the proof. □

## 4.2 Correlation Clustering on complete graphs

In this section, we present our algorithm for Correlation Clustering on complete graphs and its analysis. Our algorithm achieves an approximation ratio of 5 and is an improvement over the approximation ratio of 7 by Charikar, Gupta, and Schwartz [2017].

### 4.2.1 Summary of the algorithm

Our algorithm is based on rounding an optimal solution to the convex relaxation (P). Recall that for complete graphs, we can get a simpler relaxation by removing the variables $z$ in our convex programming formulation. We start with considering the entire vertex set of unclustered vertices. At each step $t$ of the algorithm, we select a subset of vertices as a cluster $C_t$ and remove it from the set of unclustered vertices. Thus, each vertex is assigned to a cluster exactly once and is never removed from a cluster once it is assigned.

For each vertex $w \in V$, let $\operatorname{Ball}(w, \rho) = \{u \in V : x_{uw} \leq \rho\}$ be the set of vertices within a distance of $\rho$ from $w$. For $r = 1/5$ the quantity $r - x_{uw}$ where $u \in \operatorname{Ball}(w, r)$ represents the distance from $u$ to the boundary of the ball of radius $1/5$ around $w$. Let $V_t \subseteq V$ be the set of unclustered vertices at step $t$, and define

$$L_t(w) = \sum_{u \in \operatorname{Ball}(w, r) \cap V_t} r - x_{uw}.$$

At each step $t$, we select the vertex $w_t$ that maximizes the quantity $L_t(w)$ over all unclustered vertices $w \in V_t$ and select the set $\operatorname{Ball}(w_t, 2r)$ as a cluster. We repeat this step until all the nodes have been clustered. A complete description of the algorithm can be found in Figure B.1 (supplementary material).

### 4.2.2 Overview of the analysis

Our main result for complete graphs is the following, which proves Theorem 1.3.

**Theorem 4.3.** *Algorithm 2 is a* $5-$*approximation algorithm for Correlation Clustering on complete graphs.*

For an edge $(u, v) \in E$, let $LP(u, v)$ be the LP cost of the edge $(u, v)$: $LP(u, v) = x_{uv}$ if $(u, v) \in E^+$ and $LP(u, v) = 1 - x_{uv}$ if $(u, v) \in E^-$. Let $ALG(u, v) = \mathbb{1}((u, v)$ is in disagreement $)$.

Define

$$\operatorname{profit}(u) = \sum_{(u,v) \in E} LP(u, v) - r \sum_{(u,v) \in E} ALG(u, v),$$

where $r = 1/5$. We show that for each vertex $u \in V$, we have profit$(u) \geq 0$ (see Lemma 4.4) and, therefore, the number of disagreeing edges incident to $u$ is upper bounded by $5y(u)$:

$$ALG(u) = \sum_{v:(u,v)\in E} ALG(u,v) \leq \frac{1}{r} \sum_{v:(u,v)\in E} LP(u,v) = 5y(u).$$

Thus, $\|ALG\|_q \leq 5\|y\|_q$ for any $q \geq 1$. Consequently, the approximation ratio of the algorithm is at most 5 for any norm $\ell_q$.

**Lemma 4.4.** *For every $u \in V$, we have profit$(u) \geq 0$.*

At each step $t$ of the algorithm, we create a new cluster $C_t$ and remove it from the graph. We also remove all edges with at least one endpoint in $C_t$. Denote this set of edges by

$$\Delta E_t = \{(u,v) : u \in C_t \text{ or } v \in C_t\}.$$

Now let

$$\text{profit}_t(u,v) = \begin{cases} LP(u,v) - rALG(u,v), & \text{if } (u,v) \in \Delta E \\ 0, & \text{otherwise} \end{cases}.$$

$$\text{profit}_t(u) = \sum_{v \in V_t} \text{profit}_t(u,v) = \sum_{(u,v)\in\Delta E_t} LP(u,v) - r \sum_{(u,v)\in\Delta E_t} ALG(u,v). \tag{3}$$

As all sets $\Delta E_t$ are disjoint, profit$(u) = \sum_t \text{profit}_t(u)$. Thus, to prove Lemma 4.4, it is sufficient to show that profit$_t(u) \geq 0$ for all $t$. Note that we only need to consider $u \in V_t$ as profit$_t(u) = 0$ for $u \notin V_t$.

Consider a step $t$ of the algorithm and vertex $u \in V_t$. Let $w = w_t$ be the center of the cluster chosen at this step. First, we show that since the diameter of the cluster $C_t$ is $4r$, for all negative edges $(u,v) \in E^-$ with $u, v \in C_t$, we can charge the cost of disagreement to the edge itself, that is, profit$_t(u,v)$ is nonnegative for $(u,v) \in E^-$ (see Lemma B.3). We then consider two cases: $x_{uw} \in [0,r] \cup [3r,1]$ and $x_{uw} \in (r,3r]$.

The former case is fairly simple since disagreeing positive edges $(u,v) \in E^+$ (with $x_{uw} \in [0,r] \cup [3r,1]$) have a "large" LP cost. In Lemma B.4 and Lemma B.5, we prove that the cost of disagreement can be charged to the edge itself and hence profit$_t(u) \geq 0$.

We then consider the latter case. For vertices $u$ with $x_{uw} \in (r,3r]$, profit$_t(u,v)$ for some disagreeing positive edges $(u,v)$ might be negative. Thus, we split the profit at step $t$ for such vertices $u$ into the profit they get from edges $(u,v)$ with $v$ in Ball$(w,r) \cap V_t$ and from edges with $v$ in $V_t \setminus \text{Ball}(w,r)$. That is,

$$\text{profit}_t(u) = \underbrace{\sum_{v \in \text{Ball}(w,r)} \text{profit}_t(u,v)}_{P_{high}(u)} + \underbrace{\sum_{v \in V_t \setminus \text{Ball}(w,r)} \text{profit}_t(u,v)}_{P_{low}(u)}.$$

Denote the first term by $P_{high}(u)$ and the second term by $P_{low}(u)$. We show that $P_{low}(u) \geq -L_t(u)$ (see Claim B.6 and Lemma B.7) and $P_{high} \geq L_t(w)$ (see Claim B.8 and Lemma B.9) and conclude that profit$_t(u) = P_{high}(u) + P_{low}(u) \geq L_t(w) - L_t(u) \geq 0$ since $L_t(w) = \max_{w' \in V_t} L_t(w') \geq L_t(u)$.

This finishes the proof of Lemma 4.4.

## 4.3 Correlation Clustering with AKS Min Max Objective

In this section, we present our improved algorithm for Correlation Clustering with AKS Min Max Objective. Our algorithm produces a clustering of cost at most $(2 + \varepsilon)OPT$, which improves upon the bound of $O(\sqrt{\log n \cdot \max\{\log |E^-|, \log(k)\}})-$approximation algorithm studied by Ahmadi, Galhotra, Khuller, Saha, and Schwartz [2019a].

For a subset $S \subseteq V$ of vertices, we use cost$^+(S)$ to refer to the weight of positive edges "associated" with $S$ that are in disagreement. These are the edges with exactly one end point in $S$. Thus, cost$^+(S) = \sum_{(u,v)\in E^+, u\in S, v\notin S} w_{uv}$. Similarly, we use cost$^-(S)$ to refer to the weight of dissimilar edges "associated" with $S$ that are in disagreement. These are the edges with both endpoints in

$S$. Thus, $\text{cost}^-(S) = \sum_{(u,v) \in E^-, u,v \in S} w_{uv}$. The total cost of the set $S$ is $\text{cost}(S) = \text{cost}^+(S) + \text{cost}^-(S)$.

Similar to the algorithm of Ahmadi et al. [2019b], our algorithm works in two phases. In the first phase, the algorithm covers all vertices of the graph with (possibly overlapping) sets $S_1, \ldots, S_k$ such that the cost of each set $S_i$ is at most $2OPT$ (i.e., $\text{cost}(S_i) \leq 2OPT$ for each $i \in \{1, \ldots, k\}$). In the second phase, the algorithm finds sets $P_1, \ldots, P_k$ such that: (1) $P_1, \ldots, P_k$ are disjoint and cover the vertex set; (2) $P_i \subseteq S_i$ (and, consequently, $\text{cost}^-(P_i) \leq \text{cost}^-(S_i)$); (3) $\text{cost}^+(P_i) \leq (1 + \varepsilon) \text{cost}^+(S_i)$.

The sets $P_1, \ldots, P_k$ are obtained from $S_1, \ldots, S_k$ using an uncrossing procedure of Bansal et al. [2011]. Hence the clustering that is output is $\mathcal{P} = (P_1, \ldots, P_k)$. The improvement in the approximation factor comes from the first phase of the algorithm.

### 4.3.1 Summary of the algorithm

At the core of our algorithm is a simple subproblem: For a given vertex $z \in V$, find a subset $S \subseteq V$ containing $z$ such that $\text{cost}(S)$ is minimized. We solve this subproblem using a linear programming relaxation, which is formulated as follows: The LP has a variable $x_u$ for each vertex $u \in V$. In the intended integral solution, we have $x_u = 1$ if $u$ is in the set $S$, and $x_u = 0$, otherwise. That is, $x_u$ is the indicator of the event "$u \in S$". The LP has only one constraint: $x_z = 1$. A complete description of the LP can be found in Figure C.1. In Claim C.1 we show that this LP is indeed a valid relaxation for our subproblem.

Moreover we prove that this LP is half-integral, please see section C.1 for details. We now present our algorithm which gives a 2-approximation to the subproblem.

**Rounding algorithm for subproblem.** We present a simple rounding algorithm. Let $x^*$ be an optimal half-integral LP solution to the problem. We obtain an integral solution $x$ by rouding down $x^*$, that is $x_u = \lfloor x_u^* \rfloor$ for all $u$. Thus, $\mu_{uv} \leq 2 \cdot \mu_{uv}^*$ and $\eta_{uv} \leq \eta_{uv}^*$ for all positive and negative edges respectively. Thus, the cost of the rounded solution $x$ is at most $2OPT$.

**Rounding algorithm for AKS Min Max Correlation Clustering.** To obtain a cover of all the vertices, we pick yet uncovered vertices $z \in V$ one by one and for each $z$, find a set $S(z)$ as described above. Then, we remove those sets $S(z)$ that are completely covered by other sets. The obtained family of sets $\mathcal{S} = \{S(z)\}$ satisfies the following properties: (1) Sets in $\mathcal{S}$ cover the entire set $V$; (2) $\text{cost}(S) \leq 2OPT$ for each $S \in \mathcal{S}$; (3) Each set $S \in \mathcal{S}$ is not covered by the other sets in $\mathcal{S}$ (that is, for each $S \in \mathcal{S}$, $S \not\subset \cup_{S' \in (\mathcal{S} \setminus \{S\})} S'$). However, sets $S$ in $\mathcal{S}$ are not necessarily disjoint.

Following Ahmadi et al. [2019b], we then apply an uncrossing procedure developed by Bansal et al. [2011] to the sets $S_i$ in $\mathcal{S}$ and obtain disjoint sets $P_i$ such that (1) $P_i \subset S_i$ and (2) $\text{cost}^+(P_i) \leq \text{cost}^+(S_i) + \varepsilon OPT$ for each $i$ (see Lemma C.3 in Section C.2). We have $\text{cost}^+(P_i) \leq \text{cost}^+(S_i) + \varepsilon OPT$ and $\text{cost}^-(P_i) \leq \text{cost}^-(S_i)$, since $P_i$ is a subset of $S_i$. Thus, $\text{cost}(P_i) \leq \text{cost}(S_i) + \varepsilon OPT$ and, consequently, $P_1, \ldots, P_k$ is a $2(1 + \varepsilon)$-approximation for Correlation Clustering with the AKS Min Max objective. We note that by slightly modifying our algorithm we can obtain a 2-approximation.

Finally, we show that AKS Min-Max Correlation Clustering is at least as hard as Vertex Cover (see C.3 for details). Vertex Cover is NP-hard to approximate within any constant factor better than 2 assuming the Unique Games conjecture (UGC) (see Khot and Regev [2008]). Thus, our algorithm gives the best possible approximation if UGC holds.

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
