[Supplementary Material]

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

The sets $P_1, \ldots, P_k$ are obtained from $S_1, \ldots, S_k$ using an uncrossing procedure of Bansal et al. [2011]. Hence the clustering that is output is $\mathcal{P} = (P_1, \ldots, P_k)$. The improvement in the approximation factor comes from the first phase of the algorithm.

### 4.3.1 Summary of the algorithm

At the core of our algorithm is a simple subproblem: For a given vertex $z \in V$, find a subset $S \subseteq V$ containing $z$ such that $\operatorname{cost}(S)$ is minimized. We solve this subproblem using a linear programming relaxation, which is formulated as follows: The LP has a variable $x_u$ for each vertex $u \in V$. In the intended integral solution, we have $x_u = 1$ if $u$ is in the set $S$, and $x_u = 0$, otherwise. That is, $x_u$ is the indicator of the event "$u \in S$". The LP has only one constraint: $x_z = 1$. A complete description of the LP can be found in Figure C.1. In Claim C.1 we show that this LP is indeed a valid relaxation for our subproblem.

Moreover we prove that this LP is half-integral, please see section C.1 for details. We now present our algorithm which gives a 2-approximation to the subproblem.

**Rounding algorithm for subproblem.** We present a simple rounding algorithm. Let $x^*$ be an optimal half-integral LP solution to the problem. We obtain an integral solution $x$ by rouding down $x^*$, that is $x_u = \lfloor x_u^* \rfloor$ for all $u$. Thus, $\mu_{uv} \le 2 \cdot \mu_{uv}^*$ and $\eta_{uv} \le \eta_{uv}^*$ for all positive and negative edges respectively. Thus, the cost of the rounded solution $x$ is at most 2OPT.

**Rounding algorithm for AKS Min Max Correlation Clustering.** To obtain a cover of all the vertices, we pick yet uncovered vertices $z \in V$ one by one and for each $z$, find a set $S(z)$ as described above. Then, we remove those sets $S(z)$ that are completely covered by other sets. The obtained family of sets $\mathcal{S} = \{S(z)\}$ satisfies the following properties: (1) Sets in $\mathcal{S}$ cover the entire set $V$; (2) $\operatorname{cost}(S) \le 2OPT$ for each $S \in \mathcal{S}$; (3) Each set $S \in \mathcal{S}$ is not covered by the other sets in $\mathcal{S}$ (that is, for each $S \in \mathcal{S}$, $S \not\subset \cup_{S' \in (\mathcal{S} \setminus \{S\})} S'$). However, sets $S$ in $\mathcal{S}$ are not necessarily disjoint.

Following Ahmadi et al. [2019b], we then apply an uncrossing procedure developed by Bansal et al. [2011] to the sets $S_i$ in $\mathcal{S}$ and obtain disjoint sets $P_i$ such that (1) $P_i \subset S_i$ and (2) $\operatorname{cost}^+(P_i) \le \operatorname{cost}^+(S_i) + \varepsilon OPT$ for each $i$ (see Lemma C.3 in Section C.2). We have $\operatorname{cost}^+(P_i) \le \operatorname{cost}^+(S_i) + \varepsilon OPT$ and $\operatorname{cost}^-(P_i) \le \operatorname{cost}^-(S_i)$, since $P_i$ is a subset of $S_i$. Thus, $\operatorname{cost}(P_i) \le \operatorname{cost}(S_i) + \varepsilon OPT$ and, consequently, $P_1, \ldots, P_k$ is a $2(1 + \varepsilon)$-approximation for Correlation Clustering with the AKS Min Max objective. We note that by slightly modifying our algorithm we can obtain a 2-approximation.

Finally, we show that AKS Min-Max Correlation Clustering is at least as hard as Vertex Cover (see C.3 for details). Vertex Cover is NP-hard to approximate within any constant factor better than 2 assuming the Unique Games conjecture (UGC) (see Khot and Regev [2008]). Thus, our algorithm gives the best possible approximation if UGC holds.

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

# A  Algorithm for Partitioning Metric Spaces

In this section, we will prove the following main theorem,

**Theorem A.1.** *There exists a polynomial-time randomized algorithm that given a metric space $(X, d)$ on $n$ points and parameter $\Delta$ returns a random partition $\mathcal{P}$ of $X$ such that the diameter of every set $P$ in $\mathcal{P}$ is at most $\Delta$ and for every $q \geq 1$ ($q \neq \infty$) and every weighted graph $G = (X, E, w)$, we have*

$$\mathbb{E}\Big[\| \operatorname{cut}(\mathcal{P}, E)\|_q\Big] \leq Cn^{\frac{q-1}{2q}} \log^{\frac{q+1}{2q}} n \cdot \Big[\Big(\sum_{u \in X} \sum_{v:(u,v) \in E} w_{uv}^q \frac{d(u,v)}{\Delta}\Big)^{1/q} +$$

$$+ \Big(\sum_{u \in X} \Big(\sum_{v:(u,v) \in E} w_{uv} \frac{d(u,v)}{\Delta}\Big)^q\Big)^{1/q}\Big], \quad (4)$$

*for some absolute constant $C$.*

We remark that our algorithm also works for $q = \infty$. Indeed, the behaviour of the algorithm does not depend on $q$ (in fact, $q$ is not even a part of the algorithm's input). Hence, inequality (1) holds for any $q < \infty$. In the limit as $q$ tends to infinity, we get the following result. We provide the details in the full version of the paper (see supplemental materials for details).

We will need the following definition.

**Definition 3.** *Let $(X, d)$ be a metric space. The $\varepsilon$-neighborhood of a set $S \subset X$ is the set of points at distance at most $\varepsilon$ from $S$:*

$$N_\varepsilon(S) = \{u \in X : \exists v \in S \text{ such that } d(u, v) \leq \varepsilon\}.$$

*The $\varepsilon$-neighborhood of the boundary of a partition $\mathcal{P}$ is the set of points*

$$N_\varepsilon(\partial \mathcal{P}) = \bigcup_{P \in \mathcal{P}} (N_\varepsilon(P) \setminus P) = \{u \in X : \exists v \in X \text{ s.t. } d(u, v) \leq \varepsilon \text{ and } \mathcal{P}(u) \neq \mathcal{P}(v)\}.$$

We first describe an algorithm which succeeds with probability at least $1/2$ and fails with probability at most $1/2$. If the algorithm succeeds it outputs a random partition $\mathcal{P}$ of $X$ such that the diameter of every set $P$ in $\mathcal{P}$ is at most $\Delta$ and for every $q$ and every weighted graph $G = (X, E, w)$, we have

$$\mathbb{E}\Big[\| \operatorname{cut}(\mathcal{P}, E)\|_q \mid \text{algorithm succeeds}\Big] \leq C'n^{\frac{q-1}{2q}} \log^{\frac{q+1}{2q}} n \cdot \Big(\sum_{u \in X} \sum_{v:(u,v) \in E} w_{uv}^q \frac{d(u,v)}{\Delta}\Big)^{1/q} +$$

$$\Big(\sum_{u \in X} \Big(\sum_{v:(u,v) \in E} w_{uv} \frac{d(u,v)}{\Delta}\Big)^q\Big)^{1/q}. \quad (5)$$

To obtain a valid partition with probability 1, we repeat our algorithm for at most $\lceil \log_2 n \rceil$ iterations till it succeeds and output the obtained solution. If the algorithm does not succeed after $\lceil \log_2 n \rceil$ iterations (which happens with probability at most $1/n$), we partition the graph using a simple deterministic procedure which we describe in the end of this section.

Our algorithm is based on the procedure for generating bounded padded stochastic decompositions (see Section 2). First, the algorithm picks a random padded decomposition $\mathcal{P}$ of the metric space $X$. Then, it finds the $\varepsilon$-neighborhood $N_\varepsilon(\partial \mathcal{P})$ of the boundary of $\mathcal{P}$. Finally, it outputs $\mathcal{P}$ if $|N_\varepsilon(\partial \mathcal{P})| \leq 2D\varepsilon/\Delta$ and fails otherwise. We present a pseudo-code for our algorithm in Figure A.1.

## A.1  Analysis

Our algorithm is scale invariant i.e., its output does not change if we multiply all distances in the metric space $(X, d)$ and the parameter $\Delta$ by some positive number $\lambda$. Thus, for the sake of analysis, we assume that $\Delta = 1$. Algorithm A.1 succeeds when $N_\varepsilon(\mathcal{P})$ has size at most $M$. Denote this event by $\mathcal{E}$. We first show that $\Pr(\mathcal{E}) \geq 1/2$.

**Lemma A.2.** *Algorithm A.1 succeeds with probability at least $1/2$.*

---

**Input:** metric space $(X, d)$ and parameter $\Delta > 0$.
**Output:** a random partition $\mathcal{P}$ of $X$.

1. Let $D = O(\log n)$ be the parameter from Theorem 2.1, $\varepsilon = 1/\sqrt{2Dn}$ and $M = 2D\varepsilon n/\Delta$.
2. Draw a random padded decomposition $\mathcal{P}$ of the metric space $(X, d)$ with parameter $\Delta$ using Theorem 2.1.
3. Find the neighborhood $N_\varepsilon(\partial\mathcal{P})$ of the partition boundary.
4. If $|N_\varepsilon(\mathcal{P})| \leq M$ then output $\mathcal{P}$; else fail.

---

Figure A.1: Metric decomposition algorithm.

*Proof.* Let $\bar{\mathcal{E}}$ be the complement of the event $\mathcal{E}$. We need to show that $\Pr(\bar{\mathcal{E}}) \leq 1/2$. To this end, we bound the expected size of the set $N_\varepsilon(\mathcal{P})$ using the second property of padded decompositions:

$$
\begin{aligned}
\mathbb{E}[|N_\varepsilon(\partial\mathcal{P})|] &= \sum_{u \in X} \Pr(u \in N_\varepsilon(\partial\mathcal{P})) \\
&= \sum_{u \in X} \Pr(\mathrm{Ball}(u, \varepsilon) \not\subset \mathcal{P}(u)) \\
&\leq \sum_{u \in X} D\varepsilon = D\varepsilon n.
\end{aligned}
$$

Here, we used that $u \in N_\varepsilon(\partial\mathcal{P})$ if and only if $\mathrm{Ball}(u, \varepsilon) \not\subset \mathcal{P}(u)$. Now, by Markov's inequality,

$$
\Pr(\bar{\mathcal{E}}) = \Pr(|N_\varepsilon(\partial\mathcal{P})| > \underbrace{2D\varepsilon n}_{M}) \leq \frac{D\varepsilon n}{2D\varepsilon n} = \frac{1}{2}.
$$

$\square$

Let $X_{uv}$ be the indicator of the event $\{\mathcal{P}(u) \neq \mathcal{P}(v)\}$ i.e., the event that points $u$ and $v$ are separated by the partition $\mathcal{P}$. By the second property of padded stochastic decompositions, we have $\mathbb{E}(X_{uv}) = \Pr(\mathcal{P}(u) \neq \mathcal{P}(v)) \leq D \cdot d(u, v)$. Since $\Pr(\mathcal{E}) \geq 1/2$, for each $(u, v) \in E$, we have

$$
\mathbb{E}[X_{uv} \mid \mathcal{E}] \leq \frac{\mathbb{E}[X_{uv}]}{\Pr(\mathcal{E})} \leq 2\mathbb{E}[X_{uv}] \leq 2D \cdot d(u, v).
$$

Consequently,

$$
\mathbb{E}[w_{uv}X_{uv} \mid \mathcal{E}] \leq 2D \cdot w_{uv}d(u, v) \quad \text{and} \tag{6}
$$
$$
\mathbb{E}[w_{uv}^q X_{uv}^q \mid \mathcal{E}] \leq 2D \cdot w_{uv}^q d(u, v). \tag{7}
$$

We split all edges $E$ into two groups: short edges, which we denote by $E_{\leq\varepsilon}$, and long edges, which we denote by $E_{>\varepsilon}$. Short edges are edges of length at most $\varepsilon$; long edges are edges of length greater than $\varepsilon$. Note that $\mathrm{cut}(\mathcal{P}, E) = \mathrm{cut}(\mathcal{P}, E_{\leq\varepsilon}) + \mathrm{cut}(\mathcal{P}, E_{>\varepsilon})$.

For every subset $E' \subset E$ (in particular, for $E' = E_{\leq\varepsilon}$ and $E' = E_{>\varepsilon}$), we have

$$
\mathbb{E}\Big[\| \mathrm{cut}(\mathcal{P}, E')\|_q^q | \mathcal{E}\Big] = \sum_{u \in X} \mathbb{E}\Big[\Big(\sum_{v:(u,v)\in E'} w_{uv}X_{uv}\Big)^q | \mathcal{E}\Big]. \tag{8}
$$

We separately upper bound $\mathbb{E}[\| \mathrm{cut}(\mathcal{P}, E_{\leq\varepsilon})\|_q^q \mid \mathcal{E}]$ and $\mathbb{E}[\| \mathrm{cut}(\mathcal{P}, E_{>\varepsilon})\|_q^q \mid \mathcal{E}]$ using the formula above and inequalities (6), (7) and then use the triangle inequality for $\ell_q$ norms to bound $\mathbb{E}[\| \mathrm{cut}(\mathcal{P}, E)\|_q \mid \mathcal{E}]$.

**Long edges.** Fix a vertex $u$ and consider long edges incident to $u$. Their total weight is upper bounded by

$$\sum_{v:(u,v)\in E_{>\varepsilon}} w_{uv} \leq \sum_{v:(u,v)\in E_{>\varepsilon}} w_{uv} \underbrace{\frac{d(u,v)}{\varepsilon}}_{\geq 1}.$$

Thus,

$$\Big(\sum_{v:(u,v)\in E_{>\varepsilon}} w_{uv}X_{uv}\Big)^q \leq \Big(\sum_{v:(u,v)\in E_{>\varepsilon}} w_{uv}\Big)^{q-1}\Big(\sum_{v:(u,v)\in E_{>\varepsilon}} w_{uv}X_{uv}\Big)$$

$$\leq \Big(\sum_{v:(u,v)\in E_{>\varepsilon}} \frac{w_{uv}d(u,v)}{\varepsilon}\Big)^{q-1}\Big(\sum_{v:(u,v)\in E_{>\varepsilon}} w_{uv}X_{uv}\Big).$$

Plugging this expression into formula (8) with $E' = E_{>\varepsilon}$ and using inequality (6), we get the following upper bound on $\mathbb{E}\Big[\|\operatorname{cut}(\mathcal{P}, E_{>\varepsilon})\|_q^q \mid \mathcal{E}\Big]$:

$$\sum_{u\in X}\Big(\sum_{v:(u,v)\in E_{>\varepsilon}}\frac{w_{uv}d(u,v)}{\varepsilon}\Big)^{q-1}\mathbb{E}\Big[\sum_{v:(u,v)\in E_{>\varepsilon}} w_{uv}X_{uv} \mid \mathcal{E}\Big] \leq \frac{2D}{\varepsilon^{q-1}}\sum_{u\in X}\Big(\sum_{v:(u,v)\in E_{>\varepsilon}} w_{uv}d(u,v)\Big)^q.$$

Finally, by Jensen's inequality, we have

$$\mathbb{E}\big[\|\operatorname{cut}(\mathcal{P}, E_{>\varepsilon})\|_q \mid \mathcal{E}\big] = \mathbb{E}\big[(\|\operatorname{cut}(\mathcal{P}, E_{>\varepsilon})\|_q^q)^{\frac{1}{q}} \mid \mathcal{E}\big]$$

$$\leq \Big(\mathbb{E}\big[\|\operatorname{cut}(\mathcal{P}, E_{>\varepsilon})\|_q^q \mid \mathcal{E}\big]\Big)^{\frac{1}{q}}$$

$$\leq \Big(\frac{2D}{\varepsilon^{q-1}}\sum_{u\in X}\Big(\sum_{v:(u,v)\in E_{>\varepsilon}} w_{uv}d(u,v)\Big)^q\Big)^{\frac{1}{q}}. \tag{9}$$

**Short edges.** To bound $\|\operatorname{cut}(\mathcal{P}, E_{short})\|_q$, we will make use of the following lemma.

**Lemma A.3.** *Consider non-negative (dependent) random variables $X_1, \ldots, X_n$. Suppose that at most $M$ of them are non-zero with probability 1. Then, for every $q \geq 1$, the following bound holds:*

$$\mathbb{E}\big[(X_1 + \cdots + X_n)^q\big] \leq M^{q-1}\sum_{i=1}^n \mathbb{E}\big[X_i^q\big].$$

*Proof.* Let $x_{i_1}, \ldots, x_{i_m}$ be the non-zero random variables in a certain sampling of $X_1, \ldots, X_n$ for some $m \leq M$. Suppose that $m \neq 0$. Using Jensen's inequality, we have

$$\Big(\frac{x_{i_1} + \ldots + x_{i_m}}{m}\Big)^q \leq \frac{1}{m}\sum_{j=1}^m x_{i_j}^q,$$

and, therefore,

$$\Big(x_{i_1} + \ldots + x_{i_m}\Big)^q \leq m^{q-1}\sum_{j=1}^m x_{i_j}^q \leq M^{q-1}\sum_{j=1}^m x_{i_j}^q.$$

The inequality above also holds when $m = 0$. Thus, the expectation of the left hand side is upper bounded by the expectation of the right hand side. This concludes the proof. $\qquad\square$

Fix a vertex $u$. Observe that if $(u, v)$ is a short edge which is cut by $\mathcal{P}$ then $v$ must belong to $N_\varepsilon(\partial\mathcal{P})$. Thus, the number of non-zero random variables $X_{uv}$ for a given $u$ and $(u, v) \in E_{\leq\varepsilon}$ is upper bounded by $|N_\varepsilon(\partial\mathcal{P})|$. If the algorithm succeeds, then $|N_\varepsilon(\partial\mathcal{P})| \leq M$. Thus, by Lemma A.3,

$$\mathbb{E}\Big[\Big(\sum_{v:(u,v)\in E_{\leq\varepsilon}} w_{uv}X_{uv}\Big)^q \mid \mathcal{E}\Big] \leq M^{q-1}\sum_{v:(u,v)\in E_{\leq\varepsilon}} \mathbb{E}\big[w_{uv}^q X_{uv}^q \mid \mathcal{E}\big].$$

Plugging this bound into formula (8) with $E' = E_{\leq \varepsilon}$ and using inequality (7), we get the following upper bound on $\mathbb{E}\left[\| \operatorname{cut}(\mathcal{P}, E_{\leq \varepsilon})\|_q^q \mid \mathcal{E}\right]$:

$$\sum_{u \in X} \left( M^{q-1} \sum_{v:(u,v)\in E_{\leq\varepsilon}} \mathbb{E}\left[ w_{uv}^q X_{uv}^q \mid \mathcal{E}\right] \right) \leq 2D\, M^{q-1} \sum_{u \in X} \sum_{v:(u,v)\in E_{\leq\varepsilon}} w_{uv}^q d(u,v).$$

Finally, by Jensen's inequality, we have

$$\mathbb{E}[\| \operatorname{cut}(\mathcal{P}, E_{\leq\varepsilon})\|_q \mid \mathcal{E}] \leq \left(2D\, M^{q-1}\right)^{1/q} \Big( \sum_{u \in X} \sum_{v:(u,v)\in E_{\leq\varepsilon}} w_{uv}^q d(u,v) \Big)^{1/q}. \tag{10}$$

To obtain the desired bound (5), we substitute $D = O(\log n)$, $\varepsilon = 1/\sqrt{2Dn}$, and $M = 2D\varepsilon n/\Delta$ in bounds (9) and (10) and then apply the triangle inequality for the $\ell_q$ norm.

To finish the proof of Theorem 4.2, we need to describe what we do in the unlikely event that Algorithm A.1 fails $\lceil \log_2 n \rceil$ times. In this case, we create a new graph on $X$ with edges between pairs of vertices at distance at most $1/n$ from each other and partition it into connected components. We analyze this algorithm in the full version of the paper (see supplemental materials for details).

## B    Correlation Clustering on Complete Graphs

In this section, we present our algorithm for Correlation Clustering on complete graphs and its analysis. Our algorithm achieves an approximation ratio of 5 and is an improvement over the approximation ratio of 7 by Charikar, Gupta, and Schwartz [2017].

### B.1    The Algorithm

Our algorithm is based on rounding an optimal solution to the convex relaxation (P). Recall that for complete graphs, we can get a simpler relaxation by removing the variables $z$ in our convex programming formulation. We start with considering the entire vertex set of unclustered vertices. At each step $t$ of the algorithm, we select a subset of vertices as a cluster $C_t$ and remove it from unclustered vertices. Thus, each vertex is assigned to a cluster exactly once and is never removed from a cluster once it is assigned.

For each vertex $w \in V$, let $\operatorname{Ball}(w, \rho) = \{u \in V : x_{uw} \leq \rho\}$ be the set of vertices within a distance of $\rho$ from $w$. For $r = 1/5$ the quantity $r - x_{uw}$ where $u \in Ball(w, r)$ represents the distance from $u$ to the boundary of the ball of radius $1/5$ around $w$. Let $V_t \subseteq V$ be the set of unclustered vertices at step $t$, and define

$$L_t(w) = \sum_{u \in \operatorname{Ball}(w,r) \cap V_t} r - x_{uw}.$$

At each step $t$, we select the vertex $w_t$ that maximizes the quantity $L_t(w)$ over all unclustered vertices $w \in V_t$ and select the set $Ball(w_t, 2r)$ as a cluster. We repeat this step until all the nodes have been clustered. A pseudo-code for our algorithm is given in Figure B.1.

### B.2    Analysis

In this section, we present an analysis of our algorithm.

**Theorem B.1.** *Algorithm 2 gives a 5-approximation for Correlation Clustering on complete graphs.*

For an edge $(u,v) \in E$, let $LP(u,v)$ be the LP cost of the edge $(u,v)$: $LP(u,v) = x_{uv}$ if $(u,v) \in E^+$ and $LP(u,v) = 1 - x_{uv}$ if $(u,v) \in E^-$. Let $ALG(u,v) = \mathbb{1}((u,v)$ is in disagreement $)$.

Define

$$\operatorname{profit}(u) = \sum_{(u,v)\in E} LP(u,v) - r \sum_{(u,v)\in E} ALG(u,v),$$

**Input:** Optimal solution $x$ to the linear program (P).
**Output:** Clustering $\mathcal{C}$.

1. Let $V_0 = V$, $r = 1/5$, $t = 0$.
2. **while** ($V_t \neq \varnothing$)
   - Find $w_t = \arg\max_{w \in V_t} L_t(w)$.
   - Create a cluster $C_t = \mathrm{Ball}(w_t, 2r) \cap V_t$.
   - Set $V_{t+1} = V_t \setminus C_t$ and $t = t + 1$.
3. Return $\mathcal{C} = (C_0, \ldots, C_{t-1})$.

Figure B.1: Algorithm for Correlation Clustering on complete graphs.

where $r = 1/5$. We show that for each vertex $u \in V$, we have $\mathrm{profit}(u) \geq 0$ (see Lemma B.2 below) and, therefore, the number of disagreeing edges incident to $u$ is upper bounded by $5y(u)$:

$$ALG(u) = \sum_{v:(u,v)\in E} ALG(u,v) \leq \frac{1}{r} \sum_{v:(u,v)\in E} LP(u,v) = 5y(u).$$

Thus, $\|ALG\|_q \leq 5\|y\|_q$ for any $q \geq 1$. Consequently, the approximation ratio of the algorithm is at most 5 for any norm $\ell_q$.

**Lemma B.2.** *For every $u \in V$, we have $\mathrm{profit}(u) \geq 0$.*

At each step $t$ of the algorithm, we create a new cluster $C_t$ and remove it from the graph. We also remove all edges with at least one endpoint in $C_t$. Denote this set of edges by

$$\Delta E_t = \{(u,v) : u \in C_t \text{ or } v \in C_t\}.$$

Now let

$$\mathrm{profit}_t(u,v) = \begin{cases} LP(u,v) - rALG(u,v), & \text{if } (u,v) \in \Delta E \\ 0, & \text{otherwise} \end{cases}.$$

$$\mathrm{profit}_t(u) = \sum_{v \in V_t} \mathrm{profit}_t(u,v) = \sum_{(u,v)\in\Delta E_t} LP(u,v) - r \sum_{(u,v)\in\Delta E_t} ALG(u,v). \tag{11}$$

As all sets $\Delta E_t$ are disjoint, $\mathrm{profit}(u) = \sum_t \mathrm{profit}_t(u)$. Thus, to prove Lemma B.2, it is sufficient to show that $\mathrm{profit}_t(u) \geq 0$ for all $t$. Note that we only need to consider $u \in V_t$ as $\mathrm{profit}_t(u) = 0$ for $u \notin V_t$.

Consider a step $t$ of the algorithm and vertex $u \in V_t$. Let $w = w_t$ be the center of the cluster chosen at this step. First, we show that since the diameter of the cluster $C_t$ is $4r$, for all negative edges $(u,v) \in E^-$ with $u,v \in C_t$, we can charge the cost of disagreement to the edge itself, that is, $\mathrm{profit}_t(u,v)$ is nonnegative for $(u,v) \in E^-$ (see Lemma B.3). We then consider two cases: $x_{uw} \in [0,r] \cup [3r,1]$ and $x_{uw} \in (r,3r]$.

The former case is fairly simple since disagreeing positive edges $(u,v) \in E^+$ (with $x_{uw} \in [0,r] \cup [3r,1]$) have a "large" LP cost. In Lemma B.4 and Lemma B.5, we prove that the cost of disagreement can be charged to the edge itself and hence $\mathrm{profit}_t(u) \geq 0$.

We then consider the latter case. For vertices $u$ with $x_{uw} \in (r,3r]$, $\mathrm{profit}_t(u,v)$ for some disagreeing positive edges $(u,v)$ might be negative. Thus, we split the profit at step $t$ for such vertices $u$ into the profit they get from edges $(u,v)$ with $v$ in $\mathrm{Ball}(w,r) \cap V_t$ and from edges with $v$ in $V_t \setminus \mathrm{Ball}(w,r)$. That is,

$$\mathrm{profit}_t(u) = \underbrace{\sum_{v \in \mathrm{Ball}(w,r)} \mathrm{profit}_t(u,v)}_{P_{high}(u)} + \underbrace{\sum_{v \in V_t \setminus \mathrm{Ball}(w,r)} \mathrm{profit}_t(u,v)}_{P_{low}(u)}.$$

Denote the first term by $P_{high}(u)$ and the second term by $P_{low}(u)$. We show that $P_{low}(u) \geq -L_t(u)$ (see Lemma B.9) and $P_{high} \geq L_t(w)$ (see Lemma B.7) and conclude that $\text{profit}_t(u) = P_{high}(u) + P_{low}(u) \geq L_t(w) - L_t(u) \geq 0$ since $L_t(w) = \max_{w' \in V_t} L_t(w') \geq L_t(u)$.

In the following claim, we show that we can charge the cost of disagreement of a negative edge to the edge itself.

**Claim B.3.** *For a negative edge $(u, v) \in E^-$, $\text{profit}_t(u, v)$ is always nonnegative.*

*Proof.* The only case when $(u, v)$ is in disagreement is when both $u$ and $v$ belong to the new cluster. In this case, they lie in the ball of radius $2r$ around $w$ (and thus $x_{uw}, x_{vw} \leq 2r$). Thus the distance $x_{uv}$ between them is at most $4r$ (because $x_{uv} \leq x_{uw} + x_{vw} \leq 4r$). The LP cost of the edge $(u, v)$ is at least $LP(u, v) = 1 - x_{uv} \geq 1 - 4r = r$. Thus, $\text{profit}_t(u, v) = LP(u, v) - rALG(u, v) = LP(u, v) - r \geq 0$. $\square$

In Lemma B.4 and Lemma B.5, we consider the case when $x_{uw} \in [0, r] \cup (3r, 1]$.

**Lemma B.4.** *If $x_{uw} \leq r$, then $\text{profit}_t(u, v) \geq 0$ for all $v \in V_t$.*

*Proof.* If $x_{uw} \in E^-$, then $\text{profit}_t(u, v) \geq 0$ by Claim B.3. Assume that $x_{uw} \in E^+$. Since $x_{uw} \leq r$, $u$ belongs to the cluster $C_t$. Thus, $(u, v)$ disagrees only if $v$ does not belong to that cluster. In this case, $x_{wv} \geq 2r$ and by the triangle inequality $x_{uv} \geq x_{vw} - x_{uw} \geq r$. Therefore, $\text{profit}_t(u, v) = x_{u,v} - r \geq 0$. $\square$

**Lemma B.5.** *If $x_{uw} \geq 3r$, then $\text{profit}_t(u, v) \geq 0$ for all $v \in V_t$.*

*Proof.* As in the previous lemma, we can assume that $x_{uw} \in E^+$. If $x_{uw} \geq 3r$, then $u$ does not belong to the new cluster $C_t$. Thus, $(u, v)$ disagrees only if $v$ belongs to $C_t$. In this case, $x_{wv} \leq 2r$ and by the triangle inequality $x_{uv} \geq x_{uw} - x_{vw} \geq r$. Therefore, $\text{profit}_t(u, v) = x_{u,v} - r \geq 0$. $\square$

We next consider $u$ such that $x_{uw} \in (r, 3r]$. First, we show that the profit we obtain from every edge $(u, v)$ with $v \in \text{Ball}(w, r)$ is at least $r - x_{vw}$, regardless of whether the edge is positive or negative.

**Claim B.6.** *If $x_{uw} \in (r, 3r]$ and $v \in \text{Ball}(w, r) \cap V_t$, then $\text{profit}_t(u, v) \geq r - x_{vw}$.*

*Proof.* First consider $u$ such that $x_{uw} \in (r, 2r]$. Note that $x_{uv} \geq x_{uw} - x_{vw} \geq r - x_{vw}$. Moreover, $x_{uv} \leq x_{uw} + x_{vw} \leq 2r + x_{vw}$. Thus, if $(u, v) \in E^+$, then $\text{profit}_t(u, v) \geq r - x_{vw}$. Otherwise, $\text{profit}_t(u, v) \geq (1 - 2r - x_{vw}) - r \geq 2r - x_{vw}$.

For $u \in (2r, 3r]$, note that $x_{uv} \geq x_{uw} - x_{vw} \geq 2r - x_{vw}$. Moreover, $x_{uv} \leq x_{uw} + x_{vw} \leq 3r + x_{vw}$. Thus, if $(u, v) \in E^+$, then $\text{profit}_t(u, v) \geq (2r - x_{vw}) - r \geq r - x_{vw}$. Otherwise, $\text{profit}_t(u, v) \geq (1 - 3r - x_{vw}) \geq 2r - x_{vw}$. $\square$

Using the above claim, we can sum up the profits from all vertices $v$ in $\text{Ball}(w, r)$ and lower bound $P_{high}(u)$ as follows.

**Lemma B.7.** *If $x_{uw} \in (r, 3r]$, then $P_{high}(u) \geq L_t(w)$.*

*Proof.* By Claim B.6, we have $\text{profit}_t(u, v) \geq r - x_{vw}$ for all $v \in V_t$. Thus,

$$P_{high}(u) = \sum_{v \in \text{Ball}(w,r) \cap V_t} \text{profit}_t(u, v) \geq \sum_{v \in \text{Ball}(w,r) \cap V_t} r - x_{vw} = L_t(w).$$

$\square$

We now lower bound $P_{low}(u)$. To this end. we estimate each term $\text{profit}_t(u, v)$ in the definition of $P_{low}$.

**Claim B.8.** *If $x_{uw} \in (r, 3r]$ and $v \in V_t \setminus \text{Ball}(w, r)$, then $\text{profit}_t(u, v) \geq \min(x_{uv} - r, 0)$.*

*Proof.* By Claim B.3, if $(u, v)$ is a negative edge, then $\text{profit}_t(u, v) \geq 0$. The profit is $0$ if $x_{uv} \notin \Delta E_t$ (i.e., neither $u$ nor $v$ belong to the new cluster). So let us assume that $(u, v)$ is a positive edge in $\Delta E_t$. Then, the profit obtained from $(u, v)$ is $x_{uv}$ if $(u, v)$ is in agreement and $x_{uv} - r$ if $(u, v)$ is in disagreement. In any case, $\text{profit}_t(u, v) \geq x_{uv} - r \geq \min(x_{uv} - r, 0)$. $\square$

$$\text{minimize} \quad \sum_{(u,v) \in E^+} w_{uv}|x_u - x_v| + \sum_{(u,v) \in E^-} w_{uv}(x_u + x_v - 1)^+$$

$$\text{subject to} \quad x_z = 1$$

$$0 \leq x_u \leq 1 \text{ for all } u \in V$$

Here, we use notation $(t)^+ = \max(t, 0)$.

Figure C.1: LP relaxation for covering $z$ with a low cost set $S$.

Lemma B.9 is an immediate corollary of Claim B.8.

**Lemma B.9.** *If $x_{uw} \in (r, 3r]$, then $P_{low}(u) \geq -L_t(u)$.*

*Proof.* By Claim B.8, we have $\text{profit}_t(u, v) \geq \min(x_{uv} - r, 0)$ for all $v \in V_t$. Thus,

$$\begin{aligned}
P_{low}(u) &= \sum_{v \in V_t \setminus \text{Ball}(w,r)} \text{profit}_t(u, v) \\
&\geq \sum_{v \in V_t \setminus \text{Ball}(w,r)} \min(x_{uv} - r, 0) \\
&\overset{a}{\geq} \sum_{v \in V_t} \min(x_{uv} - r, 0) \\
&\overset{b}{=} \sum_{v \in \text{Ball}(u,r) \cap V_t} x_{uv} - r \\
&= -L(u).
\end{aligned}$$

Here we used that (a) all terms $\min(x_{uv} - r, 0)$ are nonpositive, and (b) $\min(x_{uv} - r, 0) = 0$ if $v \notin \text{Ball}(u, r)$. $\qquad\square$

This finishes the proof of Lemma B.2.

## C   Correlation Clustering with the AKS Objective

Note that, after linearizing the objective function in C.1, we get the LP in C.2.

**Claim C.1.** *The LP relaxation described in Figure C.2 is a valid relaxation for the subproblem.*

*Proof.* Let us verify that this is a valid relaxation for the problem. As we discussed above, in the intended integral solution, we have $x_u = 1$ if $u$ is in the set $S$, and $x_u = 0$, otherwise. That is, $x_u$ is the indicator of the event "$u \in S$".

Consider a positive edge $(u, v) \in E^+$. In the integral solution, $|x_u - x_v| = 1$ if and only if one of the vertices $u$ or $v$ is in $S$ and the other one is not. In this case, the edge $(u, v)$ is in disagreement with $S$. Now, consider a negative edge $(u, v) \in E^-$. In the integral solution, $(x_u + x_v - 1)^+ = 1$ if and only if both $u$ and $v$ are in $S$. Again, in this case, the edge $(u, v)$ is in disagreement with $S$. Thus, this LP is a relaxation for our problem.

Note that we can linearize the $|\cdot|$ and $(\cdot)^+$ terms in the objective as follows. We can replace terms of the type $|x_u - x_v|$ with variables $\mu_{uv}$ and introduce the constraints $\mu_{uv} \geq (x_u - x_v)$ and $\mu_{uv} \geq (x_v - x_u)$. Similarly, we can replace terms of the type $(x_u + x_v - 1)^+$ with variables $\eta_{uv}$ and introduce the constraints $\eta_{uv} \geq (x_u + x_v - 1)$ and $\eta_{uv} \geq 0$. It is easy to see that the minimum values for the variables $\mu_{uv}$ and $\eta_{uv}$ is attained at $|x_u - x_v|$ and $(x_u + x_v - 1)^+$ respectively. $\qquad\square$

### C.1   Half-integrality of Subproblem Polytope

In this subsection, we show that the polytope of the subproblem that we consider is half-integral. First, we linearize the objective as described in the above subsection to obtain the equivalent relaxation (P-Cover-$z$).

$$\text{minimize} \quad \sum_{(u,v)\in E^+} w_{uv}\mu_{uv} + \sum_{(u,v)\in E^-} w_{uv}\eta_{uv}$$

$$\begin{aligned}
\text{subject to} \quad & x_z = 1 \\
& 0 \le x_u \le 1 \text{ for all } u \in V \\
& \mu_{uv} \ge (x_v - x_u) \\
& \mu_{uv} \ge (x_u - x_v) \\
& \eta_{uv} \ge 0 \\
& \eta_{uv} \ge (x_u + x_v - 1)
\end{aligned} \qquad \text{(P-Cover-}z\text{)}$$

Figure C.2: LP relaxation for covering $z$ with a low cost set $S$.

**Lemma C.2.** *The polytope (P-Cover-z) is half-integral.*

*Proof.* We will show that the (P-Cover-$z$) polytope is half-integral via contradiction, that is, we show that if an extreme point solution $(x, \mu, \eta)$ is not half-integral, then there exist feasible solution pairs $(x', \mu', \eta')$ and $(x'', \mu'', \eta'')$ such that $x$ $(\mu, \eta)$ can be written as a convex combination of $x'$ and $x''$ (respectively $(\mu', \mu''), (\eta', \eta'')$).

Towards a contradiction assume that an extreme point solution is not integral. Let $L = \{u : 0 < x_u < 1/2\}$ and $R = \{u : 1/2 < x_u < 1\}$. For each $u \in V$, define,

$$\Delta_1^u = \begin{cases} +\varepsilon, & \text{if } u \in L \\ -\varepsilon, & \text{if } u \in R \\ 0, & \text{otherwise} \end{cases} \qquad \Delta_2^u = \begin{cases} -\varepsilon, & \text{if } u \in L \\ +\varepsilon, & \text{if } u \in R \\ 0, & \text{otherwise} \end{cases}$$

Here $\varepsilon$ is chosen small enough such that: (1) $x_u \pm \varepsilon \in (0, 1/2)$ if $u \in L$, (2) $x_u \pm \varepsilon \in (1/2, 1)$ if $u \in R$, (3) $(x_u + x_v - 1) \pm 2\varepsilon < 0$ if $(x_u + x_v - 1) < 0$.

For each $u \in V$, we define, $x'_u = x_u + \Delta_1^u$ and $x''_u = x_u + \Delta_2^u$. Thus, from definition, we can see that $x'$ and $x''$ are feasible, more over, $x = \frac{1}{2}(x' + x'')$.

Consider $\mu'$ and $\mu''$ defined by $x'$ and $x''$ respectively. Notice that if $\mu_{uv} = 0$, then $\mu'_{uv} = \mu''_{uv} = 0$. Otherwise, without loss of generality, let $\mu_{uv} = x_u - x_v > 0$. If both $u, v \in L$ or $u, v \in R$, then $\mu'_{uv} = \mu''_{uv} = \mu_{uv}$. Else, $u \in R$ and $v \in L$, and $\mu'_{uv} = \mu_{uv} + 2\varepsilon$ and $\mu''_{uv} = \mu_{uv} - 2\varepsilon$. Thus, $\mu = \frac{1}{2}(\mu' + \mu'')$.

Consider $\eta'$ and $\eta''$ defined by $x'$ and $x''$ respectively. Parameter $\varepsilon$ was chosen such that if $\eta < 0$, then $\eta', \eta'' < 0$. Moreover, if $(x_u + x_v - 1) = 0$, then either $u \in L$ and $v \in R$ or $u \in R$ and $v \in L$, thus $\mu'_{uv} = \mu''_{uv} = 0$. Finally, if $\eta_{uv} > 0$, then $\eta_{uv} = (x_u + x_v - 1)$. If $u \in R$ and $v \in L$, then $\eta'_{uv} = \eta''_{uv} = \eta_{uv}$. Otherwise, $u, v \in R$ and $\eta'_{uv} = \eta_{uv} - 2\varepsilon$ and $\eta''_{uv} = \eta_{uv} + 2\varepsilon$. Thus, $\eta = \frac{1}{2}(\eta' + \eta'')$.

This contradicts the fact that $(x, \mu, \eta)$ is an extreme point solution. Thus, the polytope (P-Cover-$z$) is half-integral. $\qquad \square$

### C.2 Uncrossing Overlapping Sets

For completeness, we present here a proof of the following lemma from Bansal et al. [2011]. Denote by $\delta(S)$ the set of all positive edges leaving set $S$ in graph $G$. Then, $\text{cost}^+(S) = w(\delta(S))$.

**Lemma C.3** (Uncrossing argument in Bansal et al. [2011])**.** *There exists a polynomial-time algorithm that given a weighted graph $G = (V, E)$, a family of sets $S_1, \dots S_k$ that covers all vertices in $G$, and a parameter $\varepsilon = 1/poly(n)$, finds disjoint sets $P_1, \dots, P_k$ covering $V$ such that for each $i$:*

  *1. $P_i \subset S_i$; and*

  *2. $w(\delta(P_i)) \le w(\delta(S_i)) + \varepsilon \max_j w(\delta(S_j))$.*

*Proof.* Let us first describe the uncrossing algorithm from the paper Bansal et al. [2011]. Initially, the algorithm sets $P_i^0 = S_i \setminus \cup_{j<i} S_j$ for each $i \in \{1, \dots, k\}$. Then, at every step $t$, it finds a set $P_i^t$

violating the desired bound

$$w(\delta(P_i^t)) \leq w(\delta(S_i)) + \varepsilon \max_j w(\delta(S_j)) \tag{12}$$

and updates all sets as follows: $P_i^{t+1} = S_i$; and $P_j^{t+1} = P_j^t \setminus S_i$. The algorithm terminates and outputs sets $P_i^t$ when bound (12) holds for all sets $P_i^t$.

It easy to see that the following loop invariants hold at every step of the algorithm: (1) each $P_i^t$ is a subset of $S_i$; (2) sets $P_i^t$ are disjoint; (4) sets $P_i^t$ cover all vertices in $V$. It is also immediate that when or if the algorithm terminates sets $P_i^t$ satisfy (12). We only need to check that the algorithm stops in polynomial time.

Let $B = \max_j w(\delta(S_j))$. Define a potential function $\varphi(t) = \sum_{i=1}^{k} w(\delta(P_i))$. Observe that initially $\varphi(0) \leq 2 \sum_i w(\delta(S_i))$, since every edge cut by the partition $(P_1, \ldots, P_k)$ belongs to some $S_i$. Since, $w(\delta(S_i)) \leq B$ for all $i$, we have $\varphi(0) \leq 2kB$. We will show that at every step of the algorithm $\varphi(t)$ decreases by at least $2\varepsilon B$ and thus the algorithm terminates in at most $k/\varepsilon$ steps.

Consider step $t$ of the algorithm. Suppose that at this step of the algorithm, set $P_i^t$ violated the constraint and thus it was replaced by $S_i$. Write,

$$\varphi(t+1) - \varphi(t) = \Big(w(\delta(S_i)) - w(\delta(P_i^t))\Big) + \sum_{j \neq i}(w(\delta(P_i^{t+1})) - w(\delta(P_i^t)))$$

$$= \Big(w(\delta(S_i)) - w(\delta(P_i^t))\Big) + \sum_{j \neq i}(w(\delta(P_i^t \setminus S_i)) - w(\delta(P_i^t))).$$

Observe that for every two subsets of vertices $P$ and $S$ we have the following inequality:

$$\begin{aligned}
w(\delta(P \setminus S)) - w(\delta(P)) = & \Big(w(E(P \setminus S, V \setminus P)) + w(E(P \setminus S, P \cap S))\Big) \\
& - \Big(w(E(P \setminus S, V \setminus P)) + w(E(P \cap S, V \setminus P))\Big) \\
= & \ w(E(P \cap S, P \setminus S)) - w(E(P \cap S, V \setminus P)) \\
\leq & \ w(E(P \cap S, P \setminus S)) - w(E(P \cap S, S \setminus P)) \\
= & \ \Big(w(E(P \cap S, P \setminus S)) + w(E(S \setminus P, P \setminus S))\Big) \\
& - \Big(w(E(P \cap S, S \setminus P)) + w(E(P \setminus S, S \setminus P))\Big) \\
= & \ w(E(S, P \setminus S)) - w(E(P, S \setminus P)).
\end{aligned}$$

Also, note that $P_i^t \subset S_i \setminus P_j^t$ (since $P_i^t \subset S_i$ and all $P_j^t$ are disjoint). Consequently, $w(E(P_i^t, P_j^t)) \leq w(E(S_i \setminus P_j^t, P_j^t))$. Therefore,

$$\begin{aligned}
\varphi(t+1) - \varphi(t) = & \Big(w(\delta(S_i)) - w(\delta(P_i^t))\Big) + \sum_{j \neq i} w(E(S_i, P_j^t \setminus S_i)) - w(E(P_j^t, S_i \setminus P_j^t)) \\
\leq & \Big(w(\delta(S_i)) - w(\delta(P_i^t))\Big) + \sum_{j \neq i} w(E(S_i, P_j^t \setminus S_i)) - w(E(P_j^t, P_i^t)).
\end{aligned}$$

Using again that the sets $P_j^t$ partition $V$ into disjoint pieces, we get

$$\begin{aligned}
\varphi(t+1) - \varphi(t) \leq & \Big(w(\delta(S_i)) - w(\delta(P_i^t))\Big) + \sum_{j \neq i} w(E(S_i, P_j^t \setminus S_i)) - w(E(P_j^t, P_i^t)) \\
= & \Big(w(\delta(S_i)) - w(\delta(P_i^t))\Big) + w(E(S_i, \cup_{j \neq i} P_j^t \setminus S_i)) - w(E(\cup_{j \neq i} P_j^t, P_i^t)) \\
= & \Big(w(\delta(S_i)) - w(\delta(P_i^t))\Big) + \underbrace{w(E(S_i, V \setminus S_i))}_{=\delta(S_i)} - \underbrace{w(E(V \setminus P_i^t, P_i^t))}_{=\delta(P_i^t)} \\
= & \ 2\Big(w(\delta(S_i)) - w(\delta(P_i^t))\Big) \leq -2\varepsilon B.
\end{aligned}$$

This concludes the proof. $\qquad\qquad\qquad\qquad\qquad\qquad\qquad\qquad\qquad\qquad\qquad\qquad\quad \square$

## C.3 Hardness

We show that AKS problem is at least as hard as the vertex cover problem unless $P = NP$. This can be seen as follows.

**Theorem C.4.** *The AKS problem Min-Max objective is NP-Hard to approximate within a factor of $2 - \varepsilon$.*

*Proof.* We prove the above theorem by reducing the AKS problem to vertex cover.

**Reduction:** Let $G = (V, E)$ be the vertex cover instance. We reduce it to an instance of the AKS problem $G' = (V', E^+, E^-)$ as follows. We define the new vertex set $V'$ as $V' = V \cup \{z\}$ and the set of positive edges as the ones that connect $z$ to the vertices in $V$. That is, $E^+ = \{(z, u) : u \in V\}$. We assign each positive edge a weight of $1$. Finally, the set of negative edges are the edges $E$, that is, $E^- = E$. We assign each negative edge a weight of $\infty$.

$OPT_{AKS} \leq OPT_{VC}$ Let $S$ be an optimal vertex cover in $G$, that is $OPT_{VC} = |S|$. We will use $S$ to construct a partitioning $\mathcal{P}$ of the vertex set of $G'$, such that the cost of $\mathcal{P}$ under the AKS objective is $|S|$. We define the partitioning $\mathcal{P}$ as follows: For every vertex $v \in S$, we form a singleton cluster $\{v\}$. We also form a cluster $\{z\} \cup (V \setminus S)$ consisting of all the vertices in $V \setminus S$ and $z$. For every singleton cluster $\{v\}$, the only edge in disagreement is the similar edge connecting $v$ to $z$ of weight $1$. Thus, the cost of every singleton cluster is $1$.

Consider the cluster $\{z\} \cup (V \setminus S)$. There are no negative edges $(u, v)$ within this cluster, since for every edge $(u, v) \in E^- = E$, one of its endpoints must belong to the vertex cover $S$. Thus, the cost of the set $\{z\} \cup (V \setminus S)$ equals the number of positive edges which are cut which are exactly the edges connecting $z$ to $S$. Thus, $OPT_{AKS} \leq cost(\mathcal{P}) = |S| = OPT_{VC}$.

$OPT_{VC} \leq OPT_{AKS}$ Let $\mathcal{P}$ be an optimal partitioning of $V'$ of cost $OPT_{AKS}$. Consider the cluster $P \in \mathcal{P}$ containing the vertex $z$. Since $\mathcal{P}$ is an optimal partitioning, there can be no negative edges present within $P$. Thus the cost of the cluster $P$ equals the number of positive edges which are cut, which is exactly $|V'| - |P| = |V| + 1 - |P|$. We define the vertex cover for the graph $G$ to be $T = V \setminus (P \setminus \{z\})$. The set $T$ is a feasible set cover because every edge $(u, v) \in E$ is a negative edge of infinite weight and thus both vertices $u$ and $v$ cannot belong to the cluster $P$. Therefore, at least one of them – $u$ or $v$ – must belong to $T$. The cost of $T$ equals $|T| = |V| - (|P| - 1)$. Hence, $OPT_{VC} \leq cost(P) \leq OPT_{AKS}$. $\qquad\square$

# D  Integrality gap

In this section, we present an integrality gap example for the convex program (P). We describe an instance of the $\ell_q$ $s - t$ cut problem on $\Theta(n)$ vertices that has an integrality gap of $\Omega(n^{\frac{1}{2} - \frac{1}{2q}})$. In our integrality gap example, we describe a layered graph with $\Theta(n^{\frac{1}{2}})$ layers, with each layer consisting of a complete bipartite graph on $\Theta(n^{\frac{1}{2}})$ vertices. Between each layer $i$ and $i + 1$, there is a terminal $s_i$ which connects these two layers. Finally, the terminals $s$ and $t$ are located at opposite ends of this layered graph. We will observe that for any integral cut separating $s$ and $t$, there will be at least one vertex such that a large fraction of the edges incident to it are cut. We will show that there is a corresponding fractional solution that is cheaper compared to any integral cut as the fractional solution can "spread" the cut equally across the layers, thus not penalizing any individual layer too harshly. In doing so, we will prove the following theorem,

**Theorem D.1.** *The integrality gap for the convex relaxation (P) is $\Omega(n^{\frac{1}{2} - \frac{1}{2q}})$.*

*Proof.* We now give a more formal description of the layered graph discussed above. The construction has two parameters $a$ and $b$, so we will call such a graph $G_{a,b}$. The graph consists of $b$ layers with each layer consisting of the complete bipartite graph $K_{a,a}$. We refer to layer $i$ of the graph as $G_{a,b}^i$ and refer to the left and right hand of the bipartition as $L(G_{a,b}^i)$ and $R(G_{a,b}^i)$ respectively. In addition to these layers, the graph consists of $b + 1$ terminals $\{s, t, s_1, \ldots, s_{b-1}\}$ (we will refer to $s$ as $s_0$ and $t$ as $s_b$ interchangeably). For each $i \in \{1, \ldots, b - 1\}$, the vertex $s_i$ is connected to all the vertices in $R(G_{a,b}^i)$ and $L(G_{a,b}^{i+1})$. Finally, $s$ is connected to all the vertices in $L(G_{a,b}^1)$ and $t$ is connected to all the vertices in $R(G_{a,b}^b)$.

Figure D.1: Integrality gap example.

Consider any integral cut separating $s$ and $t$ in the graph $G_{a,b}$. Any such cut must disconnect at least one pair of consecutive terminals (if all pairs of consecutive terminals are connected, then $s$ is still connected to $t$). Thus let $j \in \{0, 1, \ldots, b\}$ be such that $s_{j-1}$ is disconnected from $s_j$ and consider the subgraph induced on $\{s_{j-1} \cup s_j \cup G_{a,b}^j\}$. We will show that this induced subgraph contains a vertex such that $\Omega(a^{\frac{1}{2}})$ of its incident edges are cut. Intuitively, since $s_{j-1}$ is separated from $s_j$, if the majority of the edges incident to $s_{j-1}$ and $s_j$ are not cut, then $s_{j-1}$ and $s_j$ have many neighbors in $L(G_{a,b}^j)$ and $R(G_{a,b}^j)$ respectively. As $G_{a,b}^j$ is highly connected, in order for $s_{j-1}$ to be separated from $s_j$, there must be a vertex in $G_{a,b}^j$ with many incident edges which are cut. If $cut(s_{j-1})$ or $cut(s_j)$ is at least $a/2$, then we are done. Otherwise, $s_j$ is connected to at least $a/2$ vertices in $R(G_{a,b}^j)$, so every $u$ adjacent to $s_{j-1}$ must have at least $a/2$ incident edges which are cut. Therefore, $OPT^q \geq \Omega(a^q)$.

We now present a fractional cut separating $s$ and $t$. If an edge $e$ connects $s_i$ to a vertex in $R(G_{a,b}^i)$ for some $i \in \{1, \ldots, b\}$, set the length of the edge to be $1/b$; otherwise set the edge length to be $0$. We let $x_{uv}$ be the shortest path metric in this graph. It is easy to see that such a solution is feasible. We now analyze the quality of this solution. For each $i \in \{1, \ldots, b\}$, we have $y_{s_i} = a/b$ and for each $u \in R(G_{a,b}^i)$, we have $y_u = 1/b$. Thus

$$LP^q = ab\left(\frac{1}{b}\right)^q + b\left(\frac{a}{b}\right)^q.$$

If $b > a$, then

$$LP^q \leq ab\left(\frac{1}{b}\right) + b\left(\frac{a}{b}\right) = 2a$$

and if $b > a$, then

$$LP^q \leq ab\left(\frac{1}{b}\right) + b\left(\frac{a}{b}\right)^q \leq a^q\left(a^{-(q-1)} + b^{-(q-1)}\right).$$

Setting $a = b = \Omega(n^{\frac{1}{2}})$ gives

$$\frac{OPT^q}{LP^q} = \Omega\left(n^{\frac{q}{2}-\frac{1}{2}}\right),$$

so the integrality gap is $\frac{OPT}{LP} = \Omega(n^{\frac{1}{2}-\frac{1}{2q}})$. $\qquad\square$

# E  Hardness of approximation

In this section, we prove the following hardness result.

**Theorem E.1.** *It is NP-hard to approximate the min $\ell_\infty$ s-t cut problem within a factor of $2 - \varepsilon$ for every positive $\varepsilon$.*

*Proof.* The proof follows a reduction from 3SAT. We will describe a procedure that reduces every instance of a 3CNF formula $\phi$ to a graph $G_\phi$ such that the minimum $\ell_\infty$ s-t cut for $G_\phi$ has a certain value if and only if the formula $\phi$ is satisfiable.

**Reduction from 3SAT:** Given a 3CNF instance $\phi$ with $n$ literals and $m$ clauses, we describe a graph $G_\phi$ with $(2 + 4n + 5m)$ vertices and $(6n + 8m)$ edges. We refer to the vertex and edge set of $G_\phi$ as $V(G_\phi)$ and $E(G_\phi)$. For every literal $x_i, i \in \{1, \ldots, n\}$, we have four nodes, $x_i^T$, $x_i^F$, $x_i^\dagger$ and $\bar{x}_i^\dagger$. Additionally, we have a "False" and a "True" node. For every $i \in \{1, \ldots, n\}$, we connect "False" with $x_i^F$ and "True" with $x_i^T$ using an infinite weight edge. Both $x_i^F$ and $x_i^T$ are connected to $x_i^\dagger$ and $\bar{x}_i^\dagger$ using edges of weight 1.

For every clause $C$ in $\phi$, we will create a gadget in $G_\phi$ consisting of five nodes. We will refer to the subgraph induced by these nodes as $G_\phi[C]$. Let the clause $C = (y_1 \vee y_2 \vee y_3)$. We have a node in the gadget for each $y_i, i \in \{1, 2, 3\}$, and two additional nodes $C_a$ and $C_b$. We connect $y_2$ and $y_3$ to $C_b$, and $y_1$ and $C_b$ to $C_a$, all using unit weight edges.

We connect the gadget $G_\phi[C]$ for clause $C = (y_1 \vee y_2 \vee y_3)$ to the main graph as follows. For each $i \in \{1, 2, 3\}$, connect the vertex for the literal $y_i$ to the vertex $y_i^\dagger$ with a unit weight edge. Finally connect the node $C_a$ to the "True" vertex using an infinite weight edge. An example of a 3CNF formula $\phi$ and the corresponding $G_\phi$ is given in Figure E.1.

**Fact 1.** Consider the gadget $G_\phi[C]$ for the clause $C = (y_1 \vee y_2 \vee y_3)$. If all three nodes $y_1, y_2$, and $y_3$ need to be disconnected from $C_a$, then either $|\operatorname{cut}_{C_a}| = 2$ or $\operatorname{cut}_{C_b} = 2$. If at most two of the three nodes $y_1, y_2$ and $y_3$ need to be disconnected from $C_a$, then there is a cut that separates those nodes from $C_a$ such that both $\operatorname{cut}_{C_a}$ and $\operatorname{cut}_{C_b}$ are at most 1.

**Lemma E.2.** *Given a 3CNF formula $\phi$, consider the graph $G_\phi$ constructed according to the reduction described above. The formula $\phi$ is satisfiable if and only if the minimum $\ell_\infty$ True-False cut $\mathcal{P}$ for the graph $G_\phi$ has value 1, that is, $\| \operatorname{cut}_{\mathcal{P}} \|_\infty = 1$.*

*Proof.* **3SAT $\Rightarrow$ minimum $\ell_\infty$ True-False cut**: If the 3CNF formula $\phi$ is satisfiable, then the graph $G_\phi$ has a minimum $\ell_\infty$ s-t cut of value exactly 1. This can be seen as follows. Given a satisfying assignment $x^*$, we will construct a cut $E_{\mathcal{P}}$ (and corresponding partition $\mathcal{P}$) such that for every vertex $u \in V(G_\phi)$, $\operatorname{cut}_{\mathcal{P}}(u) \le 1$. For every $i \in \{1, \ldots, n\}$, if $x_i^*$ is True, then include $(x_i^\dagger, x_i^F)$ and $(\bar{x}_i^\dagger, x_i^T)$ as part of the cut $E_{\mathcal{P}}$, else include $(x_i^\dagger, x_i^T)$ and $(\bar{x}_i^\dagger, x_i^F)$ as part of the cut $E_{\mathcal{P}}$. Note that this cuts exactly one edge incident to each vertex $x_i^\dagger, x_i^F, \bar{x}_i^\dagger$ and $x_i^T$ for $i \in \{1, \ldots, n\}$. Since $\phi$ has a satisfiable assignment, each clause $C$ has at least one literal which is True, and hence the node corresponding to this literal is not connected to the vertex False in $G_\phi - E_{\mathcal{P}}$. Thus, each clause $C$ has at most two literals that are False, and thus there are at most two False-True paths that go through this gadget. From Fact 1, we can know that we can include edges from $E(G_\phi[C])$ in $E_{\mathcal{P}}$ such that both $\operatorname{cut}_{\mathcal{P}}(C_a)$ and $\operatorname{cut}_{\mathcal{P}}(C_b)$ are at most 1 and the False-True paths through this gadget are disconnected. Thus, cut $E_{\mathcal{P}}$ disconnects True from False such that $\| \operatorname{cut}_{\mathcal{P}}(G_\phi) \|_\infty = 1$.

**minimum $\ell_\infty$ True-False cut $\Rightarrow$ 3SAT**: Let $G_\phi$ be the graph constructed for the 3CNF formula $\phi$ such that there is a cut $E_{\mathcal{P}} \subseteq E(G_\phi)$ (and corresponding partition $\mathcal{P}$) such that $\mathcal{P}$ separates True from False and $\| \operatorname{cut}_{\mathcal{P}}(G_\phi) \|_\infty = 1$. We will construct a satisfying assignment $x^*$ from the formula $\phi$. Since $\operatorname{cut}_{\mathcal{P}}(u) \le 1$ for every $u \in V(G_\phi)$, none of the $(True, x_i^T)$, $(x_i^F, False)$ edges are part of the cut $\mathcal{P}$ for $i \in \{1, \ldots, n\}$. In order for True to be separated from False, either the edges $(x_i^\dagger, x_i^F)$ and $(\bar{x}_i^\dagger, x_i^T)$ are part of the cut $E_{\mathcal{P}}$, or the edges $(x_i^\dagger, x_i^T)$ and $(\bar{x}_i^\dagger, x_i^F)$ are part of the cut $E_{\mathcal{P}}$. This gives us our assignment; for each $i \in \{1, \ldots, n\}$, if $(x_i^T, x_i^\dagger) \in E \setminus E_{\mathcal{P}}$, then assign $x_i^*$ as True and $\bar{x}^*$ as False. Otherwise $(x_i^F, x_i^\dagger) \in E \setminus E_{\mathcal{P}}$, so assign $x_i^*$ as False and $\bar{x}^*$ as True. Now, we show that $x^*$ is a satisfiable assignment for $\phi$. To see this, note that for each clause $C$, there exists at least one

Figure E.1: $G_\phi$ for the 3CNF formula $\phi = (x_1 \vee \bar{x}_2 \vee x_3) \wedge (x_2 \vee \bar{x}_4 \vee \bar{x}_5) \wedge (\bar{x}_1 \vee x_4 \vee x_5)$.

literal $y_i$ such that the node corresponding to $y_i$ is still connected to $C_a$. As the cut $E_\mathcal{P}$ separates True and False, $(y_i^\dagger, y_i^T) \in E \setminus E(G_\phi)$ and hence $y_i^* =$ True. Thus, the assignment $x^*$ is satisfiable for $\phi$.

$\square$

Thus, we can conclude Theorem 5.1 from the reduction procedure provided and Lemma 5.2.     $\square$

# F   Correlation Clustering on Complete Bipartite Graphs

Let $(V = L \cup R, E)$ be a complete bipartite graph with $L$ and $R$ being the bipartition of the vertex set. In this section, we provide and analyze an algorithm for correlation clustering on complete graphs

**Input:** Optimal solution $x$ to the linear program (P).
**Output:** Clustering $\mathcal{C}$.

1. Let $V_0 = V$, $r = 1/5$, $t = 0$.
2. **while** ($V_t \cap L \neq \varnothing$)
   - Find $w_t = \underset{w \in L}{\arg\max}\, L_t^R(w)$.
   - Create a cluster $C_t = \mathrm{Ball}_{L \cup R}(w_t, 2r) \cap V_t$.
   - Set $V_{t+1} = V_t \setminus C_t$ and $t = t + 1$.
3. Let $\mathcal{C}_L = (C_0, \ldots, C_{t-1})$.
4. **if** ($R \cap V_t \neq \emptyset$)
   - Let $C_R = R \cap V_t$.
5. Return $\mathcal{C} = \mathcal{C}_L \cup \{C_R\}$.

Figure F.1: Algorithm for Correlation Clustering on complete bipartite graphs.

with an approximation guarantee of 5 for minimizing the mistakes on one side of the bipartition (which without loss of generality will be $L$). The algorithm and analysis for complete bipartite graphs is very similar to the algorithm and analysis for complete graphs. At each step $t$ of our algorithm, we select a cluster center $w_t \in L$ and a cluster $C_t \subseteq (L \cup R)$ and remove it from the graph. This clustering step is repeated until all vertices in $L$ are part of some cluster. If there are any remaining vertices in $R$ which are unclustered, we put them in a single cluster.

Similar to the definition of $\mathrm{Ball}(w, \rho)$ in Section B, we define $\mathrm{Ball}_S(w, \rho) = \{u \in S : x_{uw} \leq \rho\}$. We select the cluster centers $w_t$ in step $t$ as follows. Let $V_t \subseteq V$ be the set of unclustered vertices at the start of step $t$. We redefine $L_t^S(w) = \sum_{u \in Ball_{V_t \cap S}(w, r)} r - x_{uw}$. We select $w_t$ as the vertex $w \in L$ that maximizes $L_t(w)$. We then select $Ball_{L \cup R}(w, 2r)$ as our cluster and repeat. A pseudocode for the above algorithm is provided in Figure F.1.

## F.1 Analysis

In this section, we present an analysis of our algorithm.

**Theorem F.1.** *Algorithm 3 gives a 5-approximation for Correlation Clustering on complete biparite graphs where disagreements are measured on only one side of the bipartition.*

The proof of this theorem is almost identical to the proof of Theorem B.1. We define $LP(u, v)$, $ALG(u, v)$, $\mathrm{profit}_t(u, v)$ for every edge $(u, v)$ and $\mathrm{profit}(u)$, $\mathrm{profit}_t(u)$ for every vertex $u$ as in Section B. We then show that for each vertex $u \in L$, we have $\mathrm{profit}(u) \geq 0$ and, therefore, the number of disagreeing edges incident to $u$ is upper bounded by $5y(u)$:

$$ALG(u) = \sum_{v:(u,v)\in E} ALG(u, v) \leq \frac{1}{r} \sum_{v:(u,v)\in E} LP(u, v) = 5y(u).$$

Thus, $\|ALG\|_q \leq 5\|y\|_q$ for any $q \geq 1$. Consequently, the approximation ratio of the algorithm is at most 5 for any norm $\ell_q$.

**Lemma F.2.** *For every $u \in L$, we have profit$(u) \geq 0$.*

As in Lemma 4.4, we need to show that $\mathrm{profit}_t(u) \geq 0$ for all $t$. Note that we only need to consider $u \in V_t \cap L$ as $\mathrm{profit}_t(u) = 0$ for $u \notin V_t$.

Consider a step $t$ of the algorithm and vertex $u \in V_t \cap L$. Let $w = w_t$ be the center of the cluster chosen at this step. First, we show that since the diameter of the cluster $C_t$ is $4r$, for all negative edges $(u, v) \in E^-$ with $u, v \in C_t$, we can charge the cost of disagreement to the edge itself, that

is, $\text{profit}_t(u,v)$ is nonnegative for $(u,v) \in E^-$ (see Lemma B.3). We then consider two cases: $x_{uw} \in [0, r] \cup [3r, 1]$ and $x_{uw} \in (r, 3r]$.

The former case is fairly simple since disagreeing positive edges $(u,v) \in E^+$ (with $x_{uw} \in [0, r] \cup [3r, 1]$) have a "large" LP cost. In Lemma B.4 and Lemma B.5, we prove that the cost of disagreement can be charged to the edge itself and hence $\text{profit}_t(u) \geq 0$.

We then consider the latter case. Similarly to Lemma 4.4, we split the profit at step $t$ for vertices $u$ with $x_{uw} \in (r, 3r]$ into the profit they get from edges $(u,v)$ with $v$ in $\text{Ball}_R(w, r) \cap V_t$ and from edges with $v$ in $V_t \setminus \text{Ball}_R(w, r)$. That is,

$\text{profit}_t(u) =$

$$= \underbrace{\sum_{v \in \text{Ball}_R(w,r) \cap V_t} \text{profit}_t(u,v)}_{P_{high}(u)} + \underbrace{\sum_{v \in V_t \setminus \text{Ball}_R(w,r)} \text{profit}_t(u,v)}_{P_{low}(u)}.$$

Denote the first term by $P_{high}(u)$ and the second term by $P_{low}(u)$. We show that $P_{low}(u) \geq -L_t^R(u)$ (see Lemma F.6 ) and $P_{high} \geq L_t^R(w) = \sum_{v \in \text{Ball}_R w, r \cap V_t} r - x_{vw}$ (see Lemma F.4 ) and conclude that $\text{profit}_t(u) = P_{high}(u) + P_{low}(u) \geq L_t^R(w) - L_t^R(u) \geq 0$ since $L_t^R(w) = \max_{w' \in V_t} L_t^R(w') \geq L_t^R(u)$.

Consider $u$ such that $x_{uw} \in (r, 3r]$. First, we show that the profit we obtain from every edge $(u,v)$ with $v \in \text{Ball}_R(w, r)$ is at least $r - x_{vw}$, regardless of whether the edge is positive or negative.

**Claim F.3.** *If $x_{uw} \in (r, 3r]$ and $v \in \text{Ball}_R(w, r) \cap V_t$, then $\text{profit}_t(u,v) \geq r - x_{vw}$.*

*Proof.* First consider $u$ such that $x_{uw} \in (r, 2r]$. Note that $x_{uv} \geq x_{uw} - x_{vw} \geq r - x_{vw}$. Moreover, $x_{uv} \leq x_{uw} + x_{vw} \leq 2r + x_{vw}$. Thus, if $(u,v) \in E^+$, then $\text{profit}_t(u,v) \geq r - x_{vw}$. Otherwise, $\text{profit}_t(u,v) \geq (1 - 2r - x_{vw}) - r \geq 2r - x_{vw}$.

For $u \in (2r, 3r]$, note that $x_{uv} \geq x_{uw} - x_{vw} \geq 2r - x_{vw}$. Moreover, $x_{uv} \leq x_{uw} + x_{vw} \leq 3r + x_{vw}$. Thus, if $(u,v) \in E^+$, then $\text{profit}_t(u,v) \geq (2r - x_{vw}) - r \geq r - x_{vw}$. Otherwise, $\text{profit}_t(u,v) \geq (1 - 3r - x_{vw}) \geq 2r - x_{vw}$. $\square$

Using the above claim, we can sum up the profits from all vertices $v$ in $\text{Ball}_R(w, r)$ and lower bound $P_{high}(u)$ as follows.

**Lemma F.4.** *If $x_{uw} \in (r, 3r]$, then $P_{high}(u) \geq L_t^R(w)$.*

*Proof.* By Claim , we have $\text{profit}_t(u,v) \geq r - x_{vw}$ for all $v \in R \cap V_t$. Thus,

$$P_{high}(u) = \sum_{v \in \text{Ball}_R(w,r) \cap V_t} \text{profit}_t(u,v)$$

$$\geq \sum_{v \in \text{Ball}_R(w,r) \cap V_t} r - x_{vw} = L_t^R(w).$$

$\square$

We now lower bound $P_{low}(u)$. To this end. we estimate each term $\text{profit}_t(u,v)$ in the definition of $P_{low}$.

**Claim F.5.** *If $x_{uw} \in (r, 3r]$ and $v \in V_t \setminus \text{Ball}_R(w, r)$, then $\text{profit}_t(u,v) \geq \min(x_{uv} - r, 0)$.*

*Proof.* By Claim B.3, if $(u,v)$ is a negative edge, then $\text{profit}_t(u,v) \geq 0$. The profit is 0 if $x_{uv} \notin \Delta E_t$ (i.e., neither $u$ nor $v$ belong to the new cluster). So let us assume that $(u,v)$ is a positive edge in $\Delta E_t$. Then, the profit obtained from $(u,v)$ is $x_{uv}$ if $(u,v)$ is in agreement and $x_{uv} - r$ if $(u,v)$ is in disagreement. In any case, $\text{profit}_t(u,v) \geq x_{uv} - r \geq \min(x_{uv} - r, 0)$. $\square$

Lemma F.6 is an immediate corollary of Claim F.5.

**Lemma F.6.** *If $x_{uw} \in (r, 3r]$, then $P_{low}(u) \geq -L_t^R(u)$.*

*Proof.* By Claim B.8, we have $\text{profit}_t(u, v) \geq \min(x_{uv} - r, 0)$ for all $v \in V_t$. Thus,

$$
\begin{aligned}
P_{low}(u) &= \sum_{v \in V_t \setminus \text{Ball}_R(w,r)} \text{profit}_t(u, v) \\
&\geq \sum_{v \in V_t \setminus \text{Ball}_R(w,r)} \min(x_{uv} - r, 0) \\
&\overset{a}{\geq} \sum_{v \in V_t} \min(x_{uv} - r, 0) \\
&\overset{b}{=} \sum_{v \in \text{Ball}_R(u,r) \cap V_t} x_{uv} - r \\
&= -L_t^R(u).
\end{aligned}
$$

Here we used that (a) all terms $\min(x_{uv} - r, 0)$ are nonpositive, and (b) $\min(x_{uv} - r, 0) = 0$ if $v \notin \text{Ball}(u, r)$. $\qquad\square$