[Reviews · NeurIPS 2019]

Reviewer 1



This paper studies a number of versions of the correlation clustering problem. One is given a (not-necessarily complete) graph G(V,E+,E-) with positive and negative edges, and seeks a partition of the vertices that minimizes the \ell_q norm of the "disagreement vector" --- that is, of the vector having in position v (for v \in V) the number of positive neighbors of v that are not in v's cluster, plus the number of negative neighbors of v that are in v's cluster. The usual correlation clustering problem minimizes the \ell_1 norm of the vector (that is, the total number of "mistakes" made by the partition). The \ell_q generalization of the correlation clustering problem was introduced by Puleo et al (2016). Currently, the best algorithms known (i) for complete graphs, general q, is a 7-approximation, (ii) for general graphs, q = \infty, is a O(\sqrt{n}) approximation. This paper gives: - an approximation algorithm for the non-complete graph version of the \ell_q generalization of the Correlation clustering problem. The authors show a O(n^{1/2 - 1/(2q)} \polylog n) approximation for each constant q --- in particular, in the case of q = 2, the authors give a O~(n^{1/4}) approximation; - moreover, for the case of complete graphs, the authors give a 5-approximation for general q; and - a better approximation algorithm for another variant of the problem introduced in Ahmadi et al. The algorithms are based on convex relaxations of the problems. The first algorithm then uses a metric space partitioning lemma to round the convex relaxation. The results are interesting to me. Minor comment: - it would be useful if the authors commented on the polynomial time (non-integral) solvability of their convex programs.

Reviewer 2



Overall this is solid theoretical work but the results are hardly too surprising. Given that the results for l_1 and l_\infty were previously known the results in this paper give a careful interpolation between the two regimes for l_q with q>=1. The paper is fairly well-written but will probably only be of interest to experts in the field. Given that no code or experiments are provided, I believe that this paper will be a much better fit for a standard TCS conference. Furthermore, I am not sure how well will presented algorithms perform on large datasets given that polynomials in the running times are quite substantial. Editorial remarks: -- Definition 1: disjoint set -> disjoint sets -- Definition 2 is very difficult to parse. What is \delta, does the property have to hold for all \delta? What is D, should it be a part of the definition as “D-separating padded decomposition” similar to what is used below in the paper. -- Would be helpful to give more intuition upfront for why \max is required in the objective. Minor remarks: -- Change “x approximation algorithm” to “x-approximation algorithm” everywhere and use hyphenation consistently throughout in other places as well (e.g. l_q-norm, polynomial-time algorithm, etc.)

Reviewer 3



The results above are achieved in a very modular way. Particularly, the core of their algorithm for Theorem 1.1 is an algorithm for decomposing metric spaces into pieces of small diameter. This is then used by first defining a metric space on the graph according to the optimal solution of a convex relaxation of the problem. Overall, Correlation Clustering is an important problem which is of special interest to the NeurIPS community. This is a strong paper that presents improved algorithms for many natural variants of the problem which were mostly studied before. The paper is also very well-written and uses many interesting ideas here and there. I would highly suggest acceptance.

[Author Response · NeurIPS 2019]

We thank the anonymous reviewers for their valuable feedback and comments. We will address all their comments and be sure to fix minor mistakes and typos in the revised version of our paper. In the paper, we present three algorithms.

The first two algorithm are for minimizing the $\ell_q$ norm of the disagreements vector on arbitrary and complete graphs. We note that both algorithms can be implemented in practice (the algorithms are not particularly complex). Both algorithms require that we first solve the convex program $(P)$. This program has a polynomial number of linear constraints, and its objective function is convex: This is because the objective function, $\max(\|y\|_q^q, \sum_u z_u)$, is the maximum of two convex functions. The first function, $\|y\|_q^q$ is the sum of $q$-th powers of the variables $y_u$ which are positive. Thus, $\|y\|_q^q$ is convex and differentiable. The second function, $\sum_u z_u$ is a linear function. Therefore, we can use off-the-shelf convex solvers (quadratic solvers for $\ell_2$) to get an optimal solution to $(P)$.

The third algorithm is for a cluster-wise local objective. The algorithm consists of solving a simple linear program for each vertex in the graph. This linear program has a $O(n^2)$ constraints and hence is relatively fast to solve.

[Meta-Review · NeurIPS 2019]

This paper presents a theoretical analysis of approximation algorithms for correlation clustering. It gives approximation algorithms for minimizing the l_q norm of the disagreement vector on arbitrary graphs and provides better approximation algorithms on complete graphs. The work is based on previous known results, which dealt with minimizing l_1 or l_inf. There is also lack of code and experiments, which makes it hard to judge how the algorithms would work on larger datasets/problems. The authors are advised to review their paper in light of the reviewers' comments.